# JIGSAW3D: DISENTANGLED 3D STYLE TRANSFER VIA PATCH SHUFFLING AND MASKING

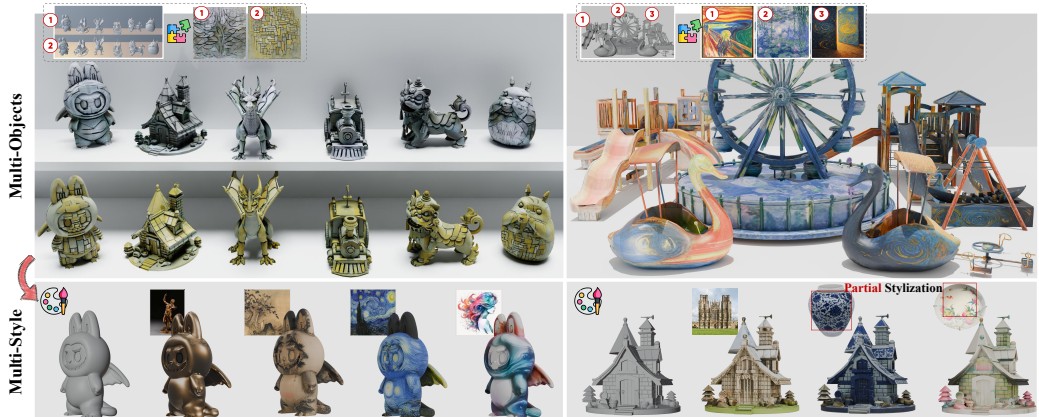

Figure 1: We propose the JIGSAW3D, a versatile 3D stylization framework that transfers stylistic statistics from 2D images to 3D meshes. Our method achieves high stylistic consistency across multiple, diverse objects in a scene (**top**). Furthermore, it demonstrates high versatility with various art styles and supports partial reference stylization for fine-grained user control (**bottom**).

## ABSTRACT

Controllable 3D style transfer seeks to restyle a 3D asset so that its textures match a reference image while preserving the integrity and multi-view consistency. The pravelent methods either rely on direct reference style token injection or score-distillation from 2D diffusion models, which incurs heavy per-scene optimization and often entangles style with semantic content. We introduce Jigsaw3D, a multi-view diffusion based pipeline that decouples style from content and enables fast, view-consistent stylization. Our key idea is to leverage the jigsaw operation—spatial shuffling and random masking of reference patches—to suppress object semantics and isolate stylistic statistics (color palettes, strokes, textures). We integrate these style cues into a multi-view diffusion model via reference-to-view cross-attention, producing view-consistent stylized renderings conditioned on the input mesh. The renders are then style-baked onto the surface to yield seamless textures. Across standard 3D stylization benchmarks, Jigsaw3D achieves high style fidelity and multi-view consistency with substantially lower latency, and generalizes to masked partial reference stylization, multi-object scene styling, and tileable texture generation.

## 1 INTRODUCTION

The field of 3D object generation has advanced rapidly, driven by progress in 3D generative modeling (Zhang et al., 2024; Xiang et al., 2025; Li et al., 2025; Wu et al., 2024; Zhang et al., 2023a), neural representations (Mildenhall et al., 2021; Park et al., 2019; Mescheder et al., 2019), and the availability of large-scale 3D datasets (Chang et al., 2015; Deitke et al., 2023b). Within this landscape, 3D stylization—transferring the artistic style of a 2D reference image to a 3D asset while preserving object identity and multi-view consistency—has emerged as a practical requirement for

virtual reality, game development, and animated content creation. Progress, however, is hampered by the absence of large-scale 3D style corpora containing paired texture and style supervision, which makes end-to-end supervised training impractical.

In response, recent approaches typically follow one of two directions. Training-free methods inject reference style cues into frozen attention layers to achieve multi-view style fusion; for example, 3D-style-LRM (Oztas et al., 2025) combines CLIP-derived features within attention modules, and Style3D (Song et al., 2024) modifies self-attention keys/values to propagate style across views. Score-distillation strategies instead leverage diffusion-based style objectives to fine-tune neural rendering pipelines (e.g., StyleTex (Xie et al., 2024)). While effective in limited settings, these paradigms often (i) struggle to disentangle style from semantic content—leading to texture leakage and degraded geometry/appearance fidelity—or (ii) require computationally expensive, per-asset optimization, limiting scalability.

To address the lack of explicit "style–texture" image pairs for training 3D stylization models, we revisit what constitutes a style reference. Conventional practice treats a "reference style image" as a natural image that entangles global semantics (object layout, parts, viewpoint) with style attributes (color palette, stroke-like texture, frequency statistics), which in turn requires substantial variation in both content and style for effective learning. We instead posit that an effective style reference should convey style independently of semantics, and that local image patches are sufficient carriers of such statistics (Wang et al., 2023). Building on this insight, we introduce a jigsaw transform that randomly shuffles and sparsely masks non-overlapping patches, destroying global structure while preserving local style cues. This enables us to synthesize style–texture supervision from textured 3D assets. Concretely, given a textured asset (e.g., from Objaverse (Deitke et al., 2023b)), we render multi-view images, apply the jigsaw transform to one rendered view to obtain a semantics-agnostic style reference, and use the remaining views as supervision targets. This procedure yields large quantities of pseudo-paired data without requiring curated style–texture pairs. We then train a multi-view stylized image generator conditioned on the style reference and geometry. The style image is encoded by a pretrained text-to-image (T2I) diffusion network; intermediate activations are extracted as disentangled style conditions. These conditions guide a multi-view diffusion model built on a U-Net backbone that integrates three complementary attention mechanisms: (a) self-attention for intra-view coherence, (b) multi-view attention to enforce cross-view consistency, and (c) reference attention that injects the style conditions to perform dynamic, style-adaptive feature recombination. In addition, geometric signals (normal and position maps) are processed by a conditional encoder and injected into the U-Net's spatial features to respect object geometry during generation. By construction, the jigsawed reference suppresses content leakage while retaining style statistics, enabling explicit content–style disentanglement and scalable training without per-asset optimization or manually paired 3D style datasets. Finally, we style-bake the multi-view outputs into a textured asset: reproject stylized views to the UV atlas with visibility/z-tests and fuse per-texel observations via seam-aware, confidence-weighted blending to obtain albedo. We then bake tangent-space normals and complete missing texels with UV-space inpainting, yielding a complete, cross-view-consistent texture.

Experiments demonstrate that our method attains state-of-the-art results on multiple 3D stylization benchmarks and generalizes to partial stylization, multi-object scene styling, and tileable texture generation, without per-asset optimization.

The main contributions of this work are summarized as follows:

- **Jigsaw-based style reference construction.** We introduce a semantics-destroying jigsaw transform—spatial patch shuffling with random masking—that disentangles style from content and synthesizes style–texture pseudo-pairs from textured 3D assets for supervised stylization training.
- **Reference-attention for stylization.** We design a trainable reference-attention module that injects disentangled style conditions to enable dynamic, style-aware feature recombination.

## 2 RELATED WORK

**3D Texture Generation.** 3D texture generation aims to create visually consistent and semantically meaningful texture maps for 3D objects. A central challenge in this task lies in achieving multi-

view geometric consistency and maintaining texture coherence across different viewpoints. Some approaches relied on optimization-based methods (Poole et al., 2022; Lin et al., 2023) that leverage pre-trained 2D diffusion models (Rombach et al., 2022) through score distillation sampling, suffering high computational costs. More related to our work are methods focusing on novel view generation with T2I models. These approaches build upon text-to-image diffusion models (Rombach et al., 2022), which provide a strong prior of 2D image appearance, and extend them to generate geometrically consistent multi-view images. Zero-1-to-3 (Liu et al., 2023a) serves as a foundational model that predicts novel views from a single image using viewpoint-conditioned diffusion. MV-Dream (Shi et al., 2023) extends this further by injecting camera parameters into the self-attention mechanism, enabling explicit 3D awareness and cross-view consistency. SyncDreamer (Liu et al., 2023b) introduces synchronized multi-view generation through feature-level fusion across views, while MV-Adapter (Huang et al., 2024) employs lightweight adapter modules to efficiently fine-tune pre-trained T2I models for multi-view synthesis. These methods commonly integrate camera embeddings or geometric constraints to maintain multi-view consistency while leveraging large-scale 3D data (Deitke et al., 2023b;a) for training.

**Image-Guided Stylization.** Image-guided stylization aims to transfer the style from a reference image to a target while preserving its semantic structure. A central challenge lies in effectively representing and transferring style features. Early approaches typically relied on statistical summarization of deep features, such as Gram matrices (Gatys et al., 2016) or channel-wise mean and variance alignment (Huang & Belongie, 2017; Lu et al., 2019). With recent advancements, research in this area has evolved along two main directions: 2D and 3D style transfer approaches.

In the domain of 2D style transfer, the rise of diffusion models has spurred the development of various fine-tuning strategies. These include full model fine-tuning (Zhang et al., 2023b; 2022), lightweight adapter-based approaches (Wang et al., 2023; Mou et al., 2024; Ye et al., 2023) that insert trainable modules into pre-trained networks, and low-rank adaptation (LoRA) (Hu et al., 2022; Frenkel et al., 2024) that captures style characteristics via weight updates. More recently, attention-based methods have attracted increasing interest. Among these, StyleAligned (Hertz et al., 2024) ensures consistent style across generated images by sharing self-attention and aligning the query and key features of target images with a reference via AdaIN (Huang & Belongie, 2017). Visual Style Prompting (Jeong et al., 2024) enables training-free style transfer by replacing the key and value features in the target's self-attention. Building on these attention designs, StyleAdapter (Wang et al., 2023) reduces semantic interference by removing feature class tokens and shuffling positional embeddings. Although the shuffling strategy in (Wang et al., 2023; Gu et al., 2018) indicates that style information can be preserved within feature patches, it has not been applied to 3D stylization tasks.

Research progress in 3D-aware stylization remains considerably limited compared to 2D stylization. Previous NeRF-based approaches (Fujiwara et al., 2024) typically depend on multi-view images and require per-asset optimization. StyleTex (Xie et al., 2024) decomposes style diffusion loss via orthogonal projection in a semantic-aware feature space, yet its test-time rendering optimization incurs significant computational overhead. Other training-free methods inject style features through attention fusion. Style3D (Song et al., 2024) directly transfers self-attention features from a 2D reference image to multi-view generation. 3D-style-LRM (Oztas et al., 2025) integrates style information through linear combinations of CLIP-based reference features.

In contrast to existing methods, we introduce a jigsaw-based disentanglement strategy to create style-texture pairs, enabling the training of dynamic style-aware feature recombination. To the best of our knowledge, our work presents the first approach to incorporate image-jigsaw for 3D stylization.

## 3 METHODS

Our method first constructs style-texture pairs from existing 3D assets to serve as training data for the framework (Sec. 3.1). Subsequently, it operates to generate multi-view stylized images and bakes these views into a stylized 3D object (Sec. 3.2, see Figure 2).

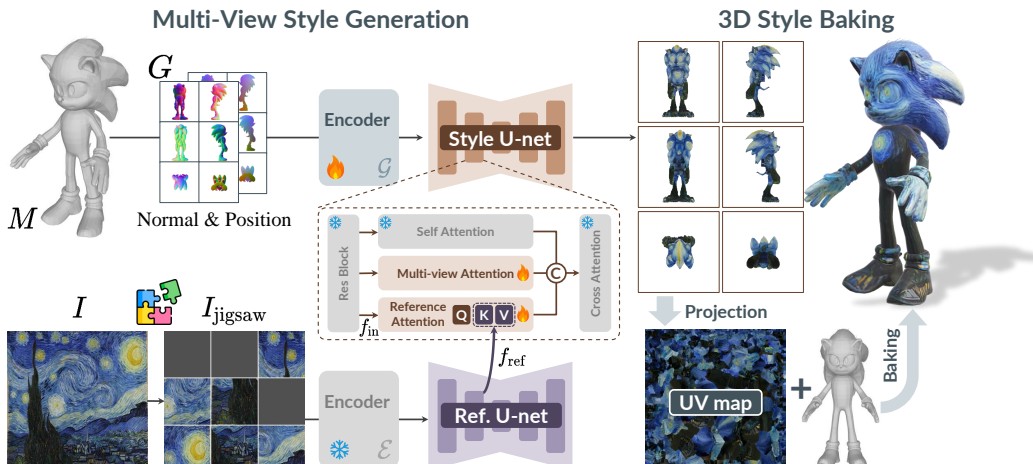

Figure 2: **Our Method Pipeline.** The whole framework contains multi-view stylized image generation and 3D style baking. **Multi-View Style Generation:** position and normal maps from the mesh $M$ are encoded and injected into a style U-Net via feature modulation, while the reference image $I$ is processed by a jigsaw operation involving image patch shuffling and random masking to extract style features. These style features are sent to a pre-trained reference U-Net to extract intermediate features that serve as keys and values in a reference attention module. Our style U-Net uses reference attention for aligning with the reference style and multi-view attention to ensure cross-view consistency. **3D Style Baking:** The generated multi-view images are projected onto the mesh's UV space, yielding a seamless UV map ready for final rendering.

## 3.1 STYLE-TEXTURE PAIRS CREATION

Unlike heavy score distillation-based approaches (Fujiwara et al., 2024; Song et al., 2024), our method adopts a data-driven manner by constructing a style-3D dataset, enabling the model to acquire stylization transfer capabilities through supervised training. However, current large-scale 3D datasets such as Objaverse (Deitke et al., 2023b) typically exhibit complex representations where semantic content and style attributes are intricately entangled within texture maps, making style extraction particularly challenging. A critical initial step involves developing an effective approach to disentangle style information from texture maps.

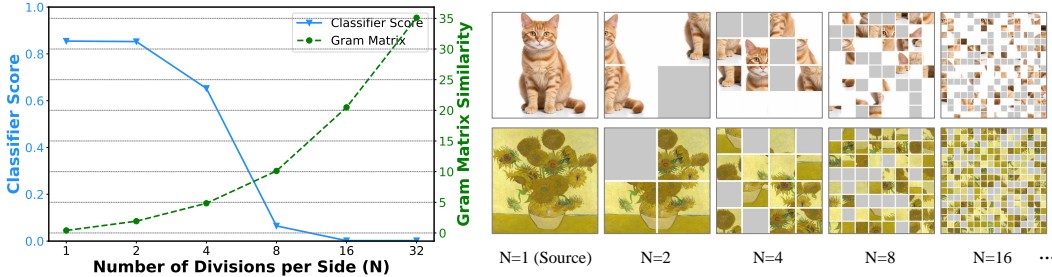

Figure 3: Analysis of style-content disentanglement through patch shuffling and masking. We apply different degrees of shuffling and a fixed mask ratio. **Left:** Quantitative evaluation of content and style attributes under increasing shuffle intensity. As $N$ (number of divisions per image side) increases, the CNN-based classification score (blue line) of shuffled images decreases sharply. At $N = 8$, semantic content is almost entirely lost. Meanwhile, the Gram matrix similarity (Gatys et al., 2016) (denoted as green dashed line) calculated between shuffled images and source images increases gradually for $N \leq 8$, indicating well-preserved style fidelity. The setting $N = 8$ strikes a good balance between semantic suppression and style preservation. **Right:** Visual examples of shuffled images using different values of $N$ and a fixed mask ratio.

**Disentanglement Motivation.** Generally disentanglement requires finding a common representation to express variations among different statistical dimensions. It has been observed in 2D style transfer that image patches can serve as effective carriers of style information (Wang et al., 2023). Based on this insight, we further posit that deliberately shuffling and masking image patches can disrupt object structures and suppress global semantics. At the same time, such patch-level shuffling preserves first- and second-order style statistics (*e.g.* mean and variance) (Huang & Belongie, 2017). This behavior is further quantitatively demonstrated in Figure 3, where beyond a certain shuffling intensity, semantic content is largely eliminated while style fidelity remains well-preserved. Motivated by these observations, we introduce a **jigsaw operation** to perform style-content disentanglement and construct our style-texture pairs.

**Jigsaw Operation.** As shown in Figure 2, for a given reference image $I \in \mathbb{R}^{C \times H \times W}$, we first partition it into a grid of non-overlapping patches $P_{i,j}$, where each patch has size $S \times S$. These patches are then shuffled using a permutation function $\sigma$, which randomly reassembles them to disrupt structural semantics:

$$I_{\text{shuffled}} = \bigcup_{i,j} P_{\sigma(i,j)}, \quad \text{where each } P_{i,j} \in \mathbb{R}^{C \times S \times S} \tag{1}$$

This shuffling operation suppresses semantic information while preserving style attributes. We further apply stochastic masking with a mask ratio $p$ to control the proportion of patches that are masked:

$$I_{\text{jigsaw}} = \text{Mask}\left(I_{\text{shuffled}}\right) = \bigcup_{i,j} \left[ M_{i,j} \odot P_{\sigma(i,j)} + (1 - M_{i,j}) \odot \mu \right] \tag{2}$$

where $M_{i,j}$ is a binary mask with elements drawn from Bernoulli$(1-p)$, and $\mu$ is the masking background value. Similar to He et al. (2022), we encourage the remaining visible regions to reconstruct the styles of the masked patches.

During training, we use the jigsaw operation to process the current 3D object from the Objaverse to create style-texture pairs. For each 3D object, we render $K$ orthogonal views as texture targets, along with several additional random views as reference images. Each reference image is processed through the jigsaw operation to obtain $I_{\text{jigsaw}}$ as model input, while the original texture targets serve as ground truth supervision. During inference, the reference image is the provided by the user. We apply shuffling operation to reference image to produce the input for stylization. Our model is trained only on the pseudo-paired dataset and remains frozen during inference.

## 3.2 Multi-view Style Generation

After processing the reference image with the jigsaw operation, we create a large style-texture pair dataset and train a multi-view style generation model using this dataset. As illustrated in Figure 2, the multi-view generation aims to produce multi-view consistent images, combining the geometric structure of $M$ and the stylistic attributes of $I$.

**Geometric information injection**. To ensure the generated multiview images retain structural information with $M$, we leverage geometric cues from $M$ and inject them into the denoising U-Net. Specifically, we first render both position and normal maps from the $M$ from $K$ predefined camera viewpoints, following the setup of Li et al. (2023) and Bensadoun et al. (2024). These maps are concatenated along the channel dimension to form the geometry condition $\mathbf{G} \in \mathbb{R}^{B \times 2K \times H \times W}$, where the $2K$ channels comprise $K$ position maps and $K$ normal maps. Then the condition $\mathbf{G}$ is processed by a trainable condition encoder $\mathcal{G}$ based on T2I-Adapter (Mou et al., 2024), which consists of a series of convolutional and downsampling layers. The resulting multi-scale features are injected directly into the corresponding scales of the Style U-Net denoiser through additive feature modulation, providing persistent geometric guidance throughout the denoising process.

**Style information injection**. To transfer style from the reference image $I$, we first apply the jigsaw operation to suppress semantic information and disentangle style features, resulting in $I_{\text{jigsaw}}$. This processed $I_{\text{jigsaw}}$ is then encoded through a pre-trained VAE encoder $\mathcal{E}$ to obtain latent features, which are fed into a pre-trained diffusion U-Net at timestep $t = 0$. We extract intermediate hidden state features $f_{\text{ref}}$ from the self-attention layers of this U-Net, which subsequently guide the stylization process in the style U-Net. Our diffusion model employs a style U-Net architecture that

incorporates a multi-branch attention block (Huang et al., 2024) after each residual layer. These blocks consist of three parallel attention mechanisms: self-attention captures contextual relationships within each view; multi-view attention enforces consistency across different viewpoints using row-wise self-attention (Li et al., 2024); and reference attention aligns the generation with the reference style by attending to $f_{\text{ref}}$, as formalized below.

**Reference Attention.** We employ cross-attention for style transfer. In this module, the original input feature map $f_{\text{in}}$ serves as the query, while $f_{\text{ref}}$ serves as both the key and value. The reference attention operation is defined as:

$$\text{RefAttention}(f_{\text{in}}, f_{\text{ref}}) = \text{softmax}\left(\frac{f_{\text{in}} f_{\text{ref}}^T}{\sqrt{d_k}}\right) f_{\text{ref}} \tag{3}$$

The softmax output represents relevance scores between the input features $f_{\text{in}}$ and style features $f_{\text{ref}}$, enabling dynamic style-aware recombination. After computing the three attention outputs, the results are summed with the original input feature $f_{\text{in}}$. The combined representation is further refined through a text-conditioned cross-attention layer. During training, ground-truth text captions with random dropout are used to improve robustness; during inference, generic prompts such as "high quality" are employed to maintain generalization and output quality.

### 3.2.1 3D STYLE BAKING

3D Style Baking projects the pre-generated multi-view stylized images onto the UV texture space to produce fully textured 3D objects. This baking process consists of three main steps: **Visibility-aware reprojection** establishes accurate pixel-to-UV correspondences while filtering occluded or invalid regions using camera and depth information; **3D inpainting** fills missing or invisible regions by computing a weighted average of the nearest neighboring pixels on the object surface. **Seamless composition** performs 2D inpainting in UV space to eliminate seam artifacts and ensure texture continuity.

## 4 EXPERIMENTS

**Implementation Details.** Our approach is built upon Stable Diffusion XL (Rombach et al., 2022). During training, we render each object from the Objaverse (Deitke et al., 2023b) to generate 6 orthogonal views as ground-truth and 4 random views as reference images. All images are scaled to a resolution of $512 \times 512$. In the jigsaw operation, the reference image is split into patches of size $64 \times 64$ during training and $128 \times 128$ during inference. A mask ratio between 0 and 0.25 achieves a balance between prediction capability and geometric consistency. For model configuration, we apply a combined conditioning dropout strategy with a probability of 0.1, which independently drops the text condition, the image condition, or both simultaneously. The model is optimized using AdamW with a learning rate of $5 \times 10^{-5}$ for 10 epochs. We employ a DDPM sampler with 50 denoising steps during inference, with classifier-free guidance scale set to 3.0. Additionally, we adjust the log-SNR offset by $\log(n)$ where $n = 6$ is the number of views.

**Evaluation Dataset.** For 3D objects, we select 20 objects from Objaverse (Deitke et al., 2023b) covering diverse categories, including both flat-surfaced and geometrically sharp shapes. Importantly, all selected meshes are distinct from those used during training to ensure a fair evaluation. For reference images, we first select style images from WikiArt (WikiArt, 2014), and additionally collect supplementary images manually from public sources. The **WikiArt dataset** includes 30 style images, with 5 examples each from 6 artistic genres: cityscape, figurative, flower painting, landscape, marina, and still-life. Furthermore, **our collected dataset** contains 40 extra images from the internet and existing publications to cover a broader spectrum of styles, such as Chinese ink painting, bronze/gold effects, Van Gogh-style art and cartoon illustrations. All images used comply with the Creative Commons Attribution 4.0 International (CC BY 4.0) license.

**Evaluation Metrics.** We employ several metrics to quantitatively evaluate performance between 6 orthogonal rendered views and the reference image. **Gram Matrix Similarity** and **AdaIN Distance** serve as style-fidelity measurements. Specifically, we extract style features from a pre-trained VGG-19 network. The Gram matrix similarity is calculated using the Frobenius norm between the style correlation matrices, and the AdaIN distance is computed as the sum of the $L_2$ norms between the

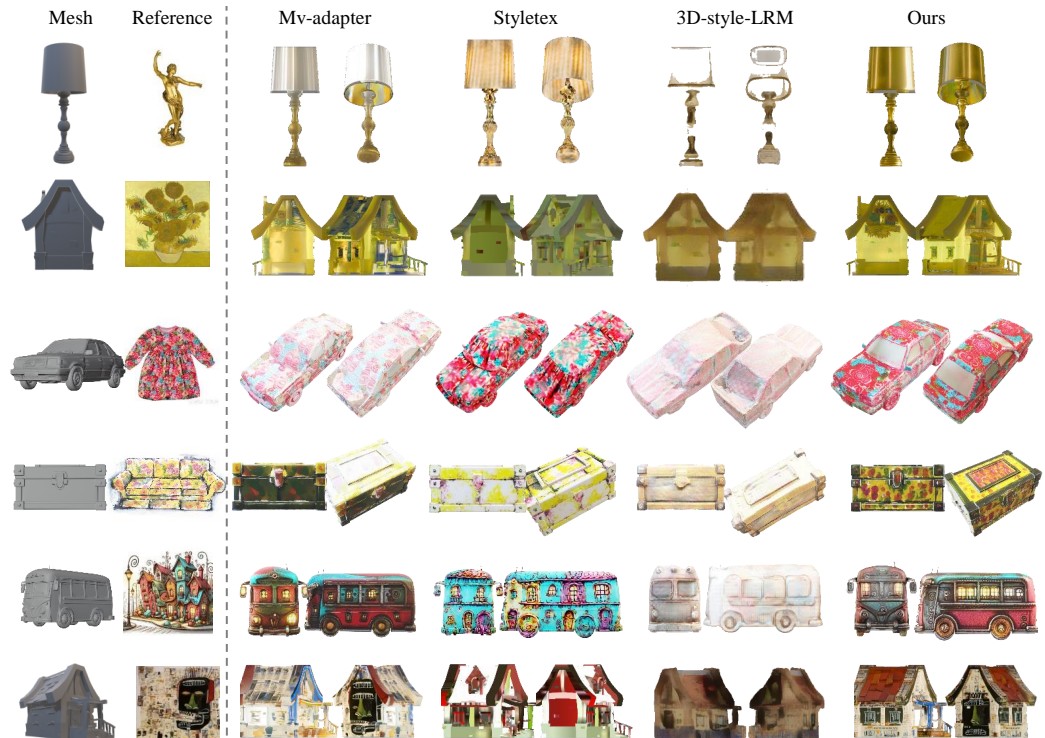

Figure 4: **Qualitative comparison between 3D stylization methods on our collected dataset and WikiArt**. The left side of the dashed line displays the input object mesh and reference image. On the right, four groups of comparative results are shown, and each group has two selected viewpoints.

feature means and standard deviations. **CLIP Score** is used to measure style-content disentanglement. A lower CLIP score indicates weaker semantic correlation and more effective disentanglement of style information from the reference. The final metrics are reported as averages across multi-view computations.

**Baseline Methods.** We compare our approach against several state-of-the-art methods in 3D texture/style generation. The selected baselines include two feed-forward generation methods (Oztas et al., 2025; Huang et al., 2024) as well as one SDS-based optimization method (Xie et al., 2024). For a fair comparison, all methods use the same reference image and object mesh. Among baselines, **3D-style-LRM** (Oztas et al., 2025) generates initial multi-view images using InstantMesh (Xu et al., 2024) and fuses style information by blending attention outputs from both the original multi-view images and the style reference within the cross-attention module. We provide the required source images that maintain strict alignment with the input mesh. **StyleTex** (Xie et al., 2024) disentangles style from reference images through orthogonal projection in the CLIP embedding space and guides texture generation iteratively using a diffusion-based style loss. **MV-Adapter** (Huang et al., 2024) employs a text-to-image diffusion model enhanced with adapter-based feature injection to produce multi-view consistent images with unified texture and style.

### 4.1 QUALITATIVE AND QUANTITATIVE COMPARISONS

**Qualitative Comparison.** Figure 4 presents a qualitative comparison of different methods for transferring reference styles onto geometric meshes. The results show that 3D-Style-LRM exhibits significant limitations in preserving the geometric fidelity of the original mesh, resulting in inconsistent surfaces. All baseline methods suffer from style infidelity compared to the reference image, such as color shifts in the clothing (row 3) or loss of texture patterns in the sofa (row 4). Additionally, MV-Adapter incorrectly transfers the entire texture map layout onto the target object. In contrast, our method demonstrates superior visual quality, with color distribution and texture details consistent with the reference. Furthermore, our disentanglement and recombination strategy effectively

| Method | Collected Data | | | WikiArt | | | Cost Time |
|---|---|---|---|---|---|---|---|
| | Gram ↓ | AdaIN ↓ | CLIP ↓ | Gram ↓ | AdaIN ↓ | CLIP ↓ | |
| StyleTex (TOG 2025) | 5.35 | 124.54 | **0.205** | 6.54 | 149.27 | **0.208** | 15min |
| MV-Adapter (ICCV 2025) | 4.85 | 114.04 | 0.214 | 4.91 | 122.19 | 0.213 | ∼40s |
| 3D-style-LRM (SIGGRAPH 2025) | 5.78 | 136.22 | 0.215 | 5.49 | 139.86 | 0.210 | ∼35s |
| **ours** | **4.81** | **113.38** | 0.213 | **4.82** | **120.54** | 0.210 | ∼40s |

Table 1: **Quantitative comparison between 3D stylization methods on our collected dataset and WikiArt.** Lower Gram and AdaIN value reflect better style fidelity, and lower CLIP score reflects more effective style-content disentanglement.

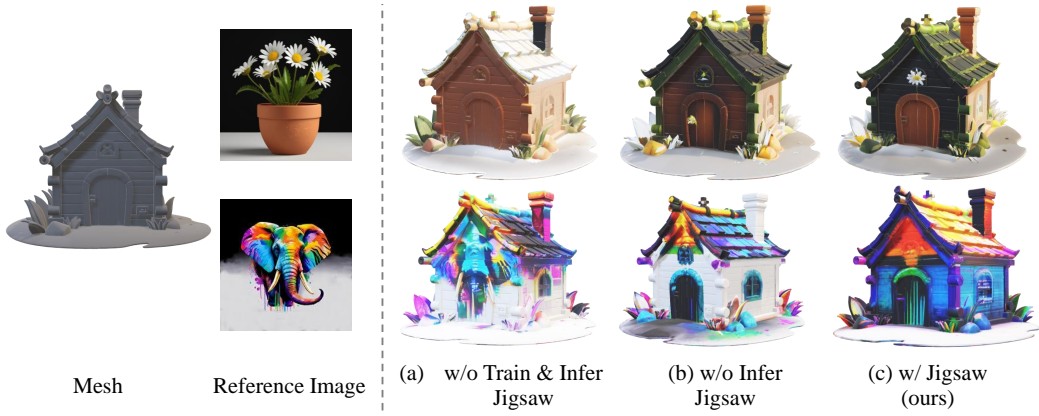

| Mesh | Reference Image | (a) w/o Train & Infer Jigsaw | (b) w/o Infer Jigsaw | (c) w/ Jigsaw (ours) |
|---|---|---|---|---|

Figure 5: **Ablation study on the Jigsaw module.** The left side shows the input object mesh and reference style image. The right side presents groups of stylization results under different Jigsaw settings: **(a) w/o Train & Infer Jigsaw**: training and reference process without jigsaw operation; **(b) w/o Infer Jigsaw**: only inference process without jigsaw operation; **(c) w/ Train & Infer Jigsaw (Ours)**: our approach applies the jigsaw operation in both training and inference phases.

applies style attributes to semantically appropriate structures. As shown in row 2, our approach applies a floral pattern to the roof of a building while maintaining pure-colored walls, enabling the assignment of distinct styles to different structural components.

**Quantitative Comparison.** Table 1 presents a quantitative comparison of 3D stylization methods. As demonstrated, our method achieves significantly superior performance on style-related metrics including Gram matrix similarity and AdaIN, indicating exceptional style consistency with the reference and multi-view coherence. Furthermore, our approach attains competitive CLIP scores, second only to StyleTex, which effectively demonstrates successful semantic disentanglement. The slightly lower CLIP performance compared to StyleTex can be attributed to the fact that StyleTex utilizes style text descriptions as additional input conditions, providing ground-truth information. Besides, compared to SDS-based methods like StyleTex, our approach is more efficient in terms of computational time.

## 4.2 ABLATION STUDY

**Ablation Study on Jigsaw Module.** Figure 5 presents an ablation study evaluating the effectiveness of the jigsaw module under different configurations. Figure (a) shows that without the jigsaw operation, the first row lacks the flower style, while the second row exhibits semantic entanglement of the elephant. When applying the jigsaw module during training as shown in Figure (b), the flower style is attributed to the house roof and the elephant semantics are compressed. Furthermore, applying jigsaw during both training and inference, as shown in Figure (c), preserves better style fidelity to the reference images, as demonstrated by the color distribution of the elephant and the detailed floral patterns on the house.

We further provide additional ablation studies, including those on the **patch size** $S$ and **mask ratio** $p$ in the jigsaw operation, in the Section A.5.

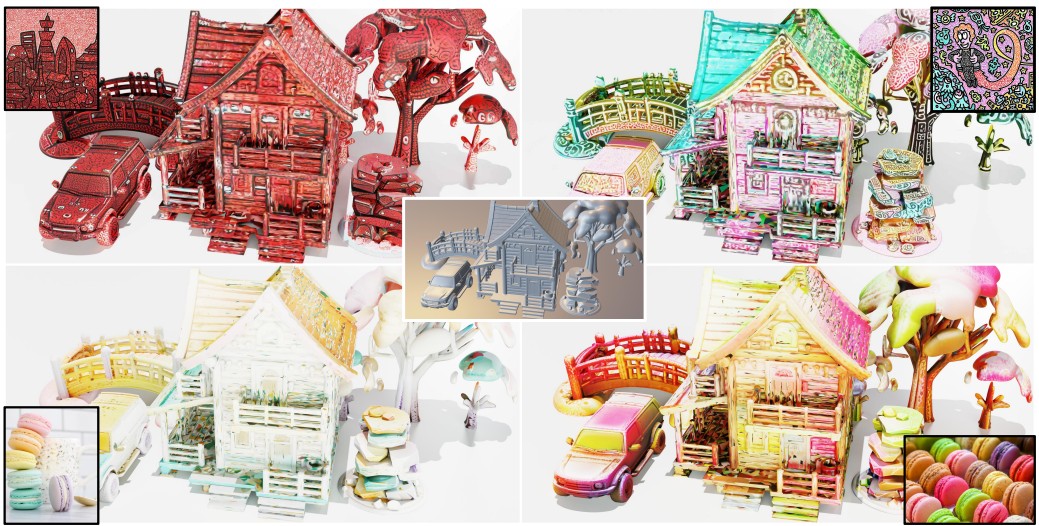

Figure 6: The first figure demonstrates strong geometric awareness, as the generated **sketch lines align precisely with the 3D feature lines of the objects**. Across all four figures, local regions also exhibit reasonable style attribution, as the house roof and wall maintain color consistency with the reference images.

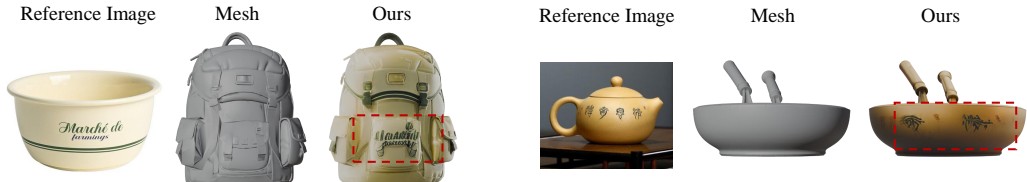

Figure 7: **Limitations of our method.** The dashed red boxes indicate failure cases in style transfer, such as text or symbolic patterns that do not match the reference image.

### 4.3 MORE APPLICATIONS AND LIMITATIONS

**Multiple Objects Scene Stylization**. We collected several scene-related 3D objects and applied our method to stylize them. As shown in Figure 6, our approach successfully achieves consistent style coherence both within individual objects and across different objects in the scene.

We further demonstrate other applications, including **tileable texture generation** and **partial stylization**, in the appendix Section A.6.

**Limitations.** During the style transfer process, our method struggles to accurately preserve fine-grained patterns such as text or symbols, as shown in Figure 7. This limitation is attributable to Stable Diffusion backbone (SDXL) we used, which lacks the capability to reliably generate or reconstruct precise textual and symbolic structures.

## 5 CONCLUSION

In this paper, we propose a new framework to transfer the style of a reference image to a 3D object while preserving its geometric structure. We propose a novel framework centered around the jigsaw operation to create a style-3D pair dataset. We apply a multiview generation pipeline to utilize the extracted style features from the jigsawed images. Geometric cues such as normal and position maps are further incorporated to enhance structural alignment. Extensive experiments demonstrate that our method achieves state-of-the-art performance across several 3D stylization benchmarks and shows strong generalization to downstream tasks such as partial object stylization, multi-object scene styling, and tileable texture generation.

**Ethical Statement.** Our work presents a method for 3D style transfer using multi-view diffusion models and a jigsaw-based disentanglement mechanism. We acknowledge that our approach could potentially be misused to create misleading or harmful content, such as generating stylized 3D objects that infringe upon intellectual property, appropriate cultural or artistic styles without permission, or propagate visual misinformation. We strongly emphasize that the use of this technology should adhere to the relevant legal frameworks and community standards. All 3D assets and reference images should be properly licensed or used in accordance with fair use principles.

**Reproducibility Statement.** To facilitate the reproducibility of our work, we provide code in the supplementary materials. A detailed README document is included, which covers the environment configuration and instructions for loading pre-trained weights.

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

# A APPENDIX

## A.1 USE OF LARGE LANGUAGE MODELS (LLMs)

We declare that Large Language Models (LLMs) were used solely as an auxiliary tool in the writing process of this paper, specifically for tasks such as checking and correcting grammatical errors and ensuring consistency in formatting and terminology. We emphasize that the core ideas, theoretical derivations, experimental design, and result analysis were entirely conceived and conducted by the authors.

## A.2 PROOF OF STYLE STATISTICS PRESERVATION UNDER SHUFFLE OPERATION

Let $I \in \mathbb{R}^{3 \times H \times W}$ denote an input image with 3 channels, height $H$, and width $W$. We partition $I$ into $N \times N$ non-overlapping patches $\{P_{i,j}\}$, each of size $\frac{H}{N} \times \frac{W}{N}$. Let $I_{\text{shuffled}}$ be the image after applying a permutation $\sigma$ to these patches:

$$I_{\text{shuffled}} = \bigcup_{i,j} P_{\sigma(i,j)}$$

**Mean Preservation.** The mean of channel $c$ of the original image is:

$$\mu_c = \frac{1}{HW} \sum_{i=1}^{H} \sum_{j=1}^{W} I(c, i, j)$$

After shuffling, only the spatial positions of pixel values are permuted. Therefore, the sum over all positions remains identical:

$$\sum_{i=1}^{H} \sum_{j=1}^{W} I(c, i, j) = \sum_{i=1}^{H} \sum_{j=1}^{W} I_{\text{shuffled}}(c, i, j)$$

Thus,

$$\mu_c' = \frac{1}{HW} \sum_{i=1}^{H} \sum_{j=1}^{W} I_{\text{shuffled}}(c, i, j) = \frac{1}{HW} \sum_{i=1}^{H} \sum_{j=1}^{W} I(c, i, j) = \mu_c$$

**Variance Preservation.** The variance of channel $c$ is defined as:

$$\sigma_c^2 = \frac{1}{HW} \sum_{i=1}^{H} \sum_{j=1}^{W} \left( I(c, i, j) - \mu_c \right)^2$$

Since both the pixel values $I(c, i, j)$ and the mean $\mu_c$ remain unchanged under shuffling, each squared term $(I(c, i, j) - \mu_c)^2$ is preserved. Therefore, the sum of squared deviations remains the same:

$$\sum_{i=1}^{H} \sum_{j=1}^{W} \left( I(c, i, j) - \mu_c \right)^2 = \sum_{i=1}^{H} \sum_{j=1}^{W} \left( I_{\text{shuffled}}(c, i, j) - \mu_c' \right)^2$$

Hence,

$$(\sigma_c^2)' = \sigma_c^2$$

According to the above, shuffling patches changes the spatial arrangement of pixels but does not alter their first-order (mean) or second-order (variance) statistics.

## A.3 STYLE CONSISTENCY METRICS.

To quantitatively evaluate style consistency between generated multi-view images and the reference style image, we employ two widely adopted perceptual style evaluation metrics: **Gram Matrix Similarity** and **AdaIN Distance**. These are considered perceptual metrics because they operate on deep feature representations rather than pixel-level values, thereby aligning with human perception

of artistic style that relies on texture patterns and statistical characteristics. Features are extracted from five key ReLU layers (1_1, 2_1, 3_1, 4_1, 5_1) of a pre-trained VGG-19 network.

**Gram Matrix Similarity** captures the correlations between feature channels. For a feature map $\mathbf{F} \in \mathbb{R}^{C \times H \times W}$, the Gram matrix $\mathbf{G} \in \mathbb{R}^{C \times C}$ is computed as:

$$\mathbf{G} = \frac{1}{C \cdot H \cdot W} \mathbf{F}\mathbf{F}^{\top}$$

The style similarity between the reference image and a generated view is measured using the Frobenius norm:

$$\mathcal{L}_{\text{Gram}} = \|\mathbf{G}_{\text{ref}} - \mathbf{G}_{\text{gen}}\|_F$$

**AdaIN Distance** measures the discrepancy in first-order (mean) and second-order (standard deviation) statistics of deep features:

$$\mathcal{L}_{\mu} = \|\mu(\mathbf{F}_{\text{ref}}) - \mu(\mathbf{F}_{\text{gen}})\|_2, \quad \mathcal{L}_{\sigma} = \|\sigma(\mathbf{F}_{\text{ref}}) - \sigma(\mathbf{F}_{\text{gen}})\|_2$$

$$\mathcal{L}_{\text{AdaIN}} = \mathcal{L}_{\mu} + \mathcal{L}_{\sigma}$$

Both metrics are computed across five VGG-19 layers and averaged over all views. Lower values indicate better style consistency. Together, these perceptual metrics provide a comprehensive assessment of style transfer quality at multiple feature levels.

## A.4 MORE RESULTS

**Further Qualitative Results of Our Methods.** We present additional multi-view rendering results of our method in Figure 8 and Figure 9. Figure 8 illustrates the outcomes of stylizing a house object using a collection of reference images with distinct artistic styles. Figure 9 extends this by applying different stylistic references to a diverse set of objects. Both sets of results highlight our method's ability to maintain strong stylistic consistency across multiple viewpoints. Besides, Figure 10 presents our method's generalization capability by applying celebrated artistic styles to amusement park scene objects. For artistic stylization, we select representative works including Monet's Water Lilies, Piet Mondrian's compositions, Edvard Munch's The Scream, Van Gogh's The Starry Night, and Ukiyo-e prints. The results show stylistic consistency and overall harmony.

**Further Qualitative Comparisons with SOTA.** Figure 11 and Figure 12 present qualitative comparisons with other methods, including Hunyuan-V2 (Zhao et al., 2025) as an additional baseline.

## A.5 MORE ABLATION STUDY

**Ablation on Mask Ratio $p$ in equation 2.** In Figure 13, we conduct an ablation study to analyze the impact of the mask ratio. The mask ratio is designed to encourage the model to learn diverse and enhanced feature representations from masked reference images. The results show employing an appropriate mask ratio can enhance the diversity of the learned feature expressions, while setting it too high (e.g., 0.75) proves detrimental. An excessive mask ratio leads to lost geometric information with generated multi-views, which in turn causes an uneven appearance with artifacts during the baking process.

**Ablation on Training and Inference Patch Size $S$ in equation 1.** The image patch serves as the primary carrier of style information, making its size a critical hyperparameter. Figure 14 showcases the qualitative rendering results of our model under different patch size configurations. A training patch size that is too small may not sufficiently contain the style information, whereas a size that is too large can cause the model to focus excessively on content details. We find that for a majority of images, a training patch size of $64 \times 64$ coupled with an inference patch size of $128 \times 128$ produces the stable results. This is likely due to the nature of the data domain, where this configuration strikes an optimal balance.

## A.6 MORE APPLICATION RESULTS

**Partial Object Stylization.** To evaluate our method's ability to understand and transfer style attributes from limited visual cues, we conduct experiments using only cropped regions of the reference image. As shown in Figure 15, our approach can successfully infer a globally consistent

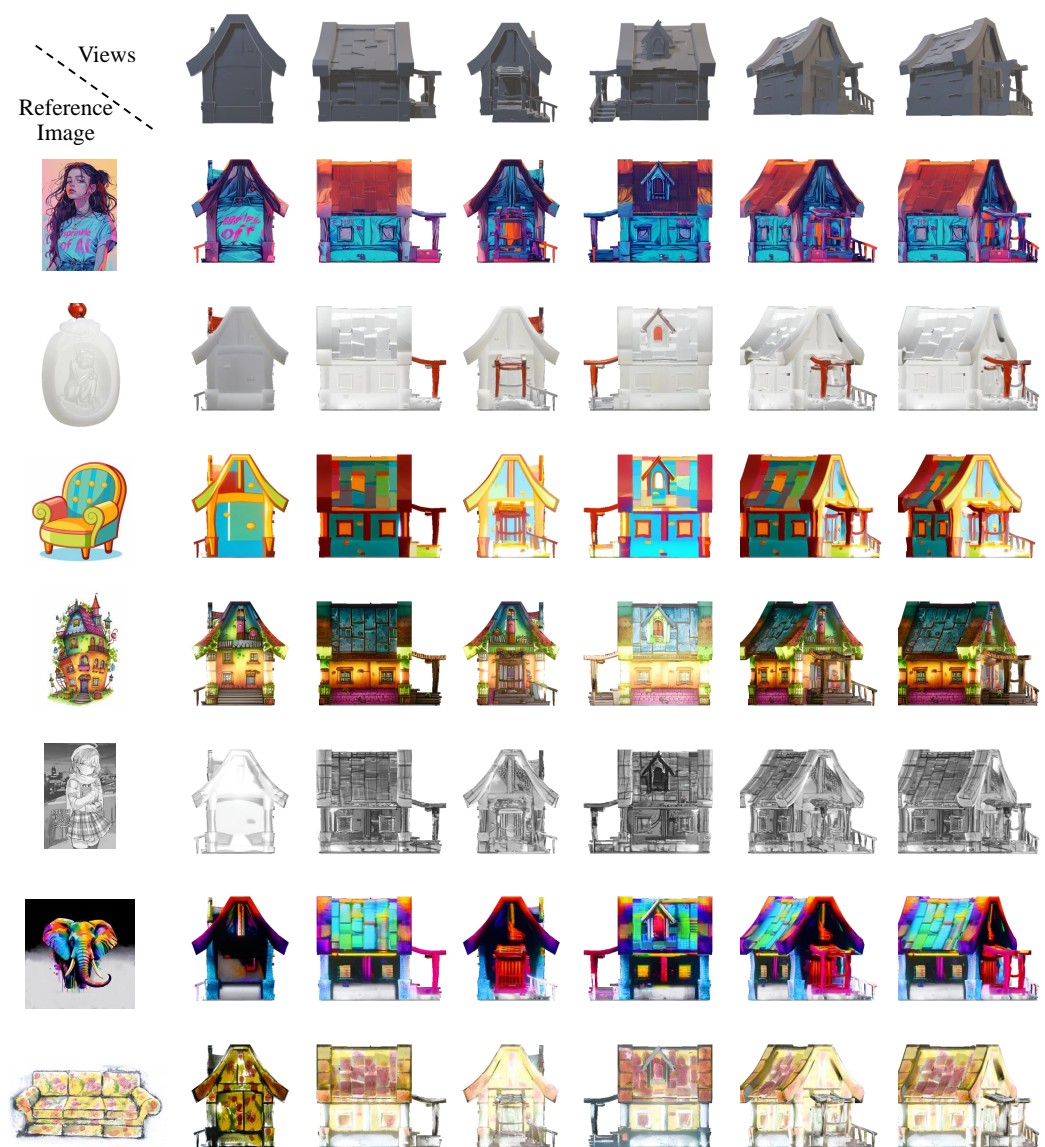

Figure 8: **More 3D Stylization Results of our method.** Multi-view rendering results by applying diverse reference style images to a "house" object.

style from a partial reference and apply it coherently to the full 3D object. This demonstrates the robustness of our model in handling partial style references while maintaining semantic and stylistic coherence.

**Tileable Texture Generation.** We further demonstrate that our method is not capable of transferring styles but also can create consistent texture tiles from an example image. As shown in Figure 16, in comparison with the direct texture mapping in Blender with noticeable irregularities and artifacts, especially around UV seams, our jigsaw-based strategy disrupts spatial biases while maintaining strong consistency and continuity across the entire 3D surface.

### A.7    VISUALIZATION ON OUR SELF-COLLECTED REFERENCE IMAGES

The visualization of our self-collected reference images is presented in Fig. 17, which includes images from the internet and existing publications to cover a broader spectrum of styles, such as Chinese ink painting, bronze and gold effects, Van Gogh-style art, and cartoon illustrations.

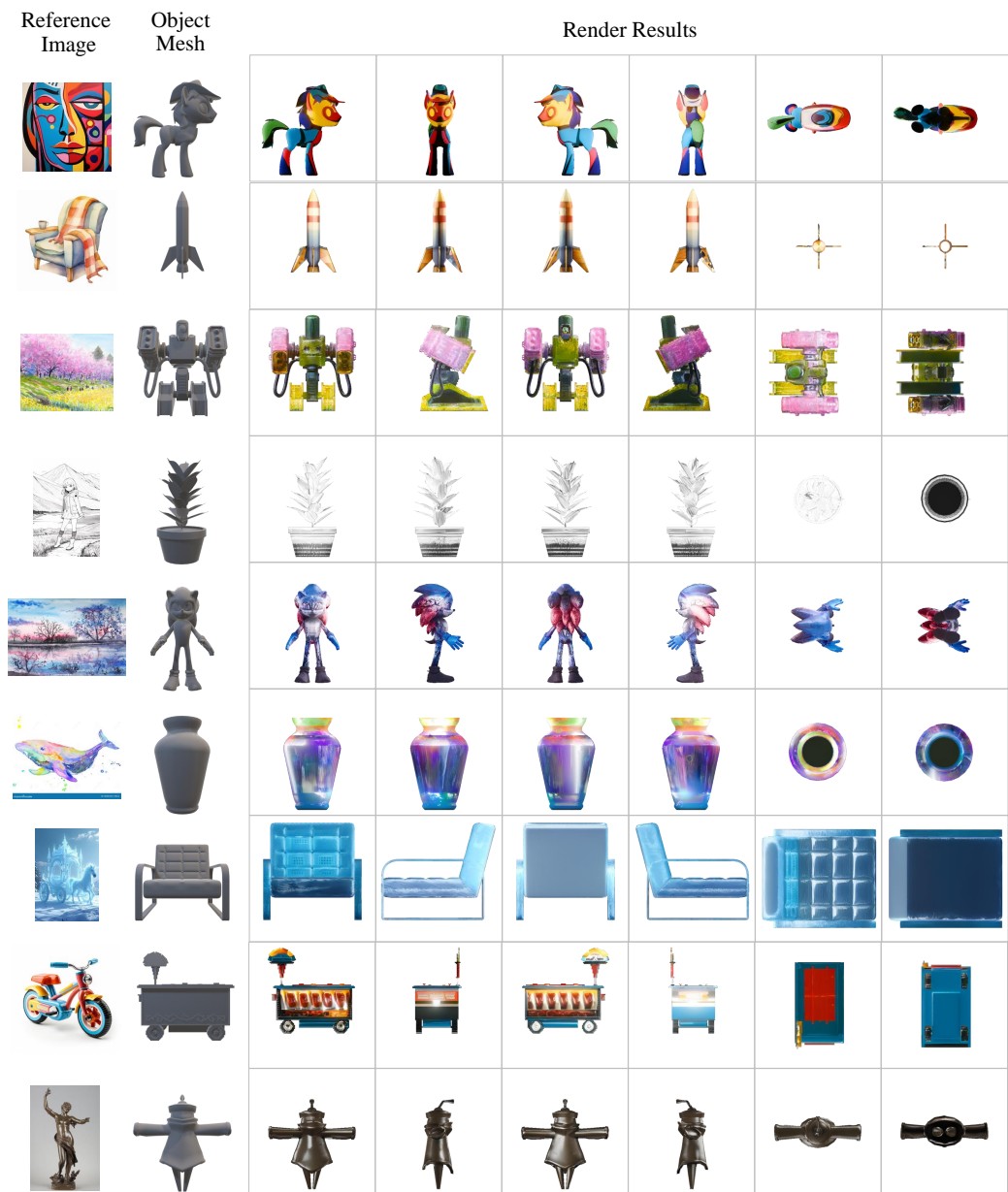

Figure 9: **More 3D Stylization Results of our method.** Multi-view rendering results by applying various reference styles to a wide range of object meshes from the Objaverse dataset.

## A.8   VISUALIZATION OF DISENTANGLEMENT RESULTS

**Disentanglement results between Jigsaw3D and MV-Adapter** are shown in Fig. 18, on flat mesh surfaces, MV-Adapter frequently overlays both the semantic content and style of the reference image onto the target geometry. In contrast, our approach effectively disentangles and transfers only the stylistic attributes, eliminating interference from irrelevant semantic elements in the reference.

## A.9   FAILURE CASES IN UV BAKING AND IMPROVEMENT STRATEGY

**Failure Cases in UV Baking and Improvement Strategy.** Fig. 19 shows the limitations of UV baking in handling occluded regions and our proposed solution. During multi-view generation, we generate 2 random viewpoints to ensure comprehensive texture coverage in the UV inpainting

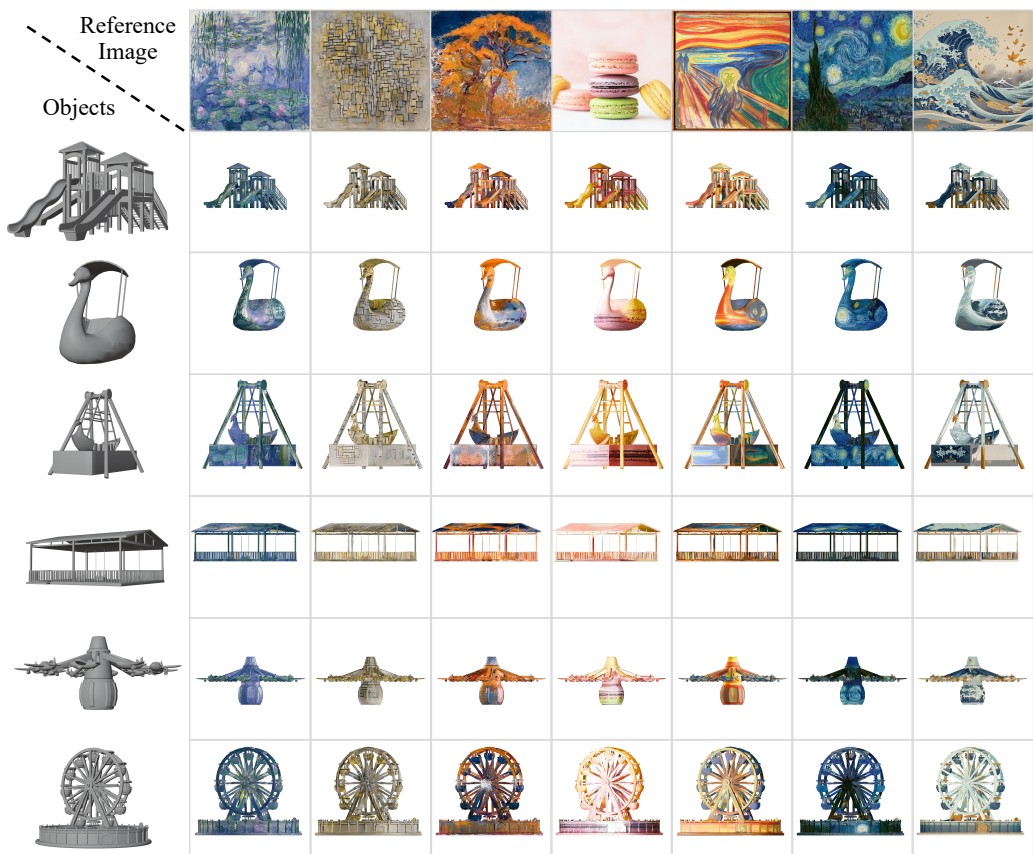

Figure 10: **Additional qualitative results of our method applied to park scene objects and artistic styles.**

process, which effectively resolves missing or incomplete texture regions. This enhancement consistently produces more coherent and complete surface textures.

### A.10 COMPARISON WITH 3DGS-BASED STYLIZATION METHODS

The comparison with 3DGS-based stylization methods is shown in Fig. 20. For our method, we first generate a scene-level mesh using Hunyuan3D (Zhao et al., 2025) with three different views as input. The mesh and reference image are then processed by our `Jigsaw3D` framework to render the output 3D asset as stylized scene images. For 3DGS stylization, we provide 120 multi-view images to StyleGaussian for Gaussian splatting reconstruction and stylization. Comparative results demonstrate that StyleGaussian fails to preserve reference texture patterns and exhibits texture incompleteness, whereas our approach achieves superior scene-style consistency and maintains coherent stylization across objects.

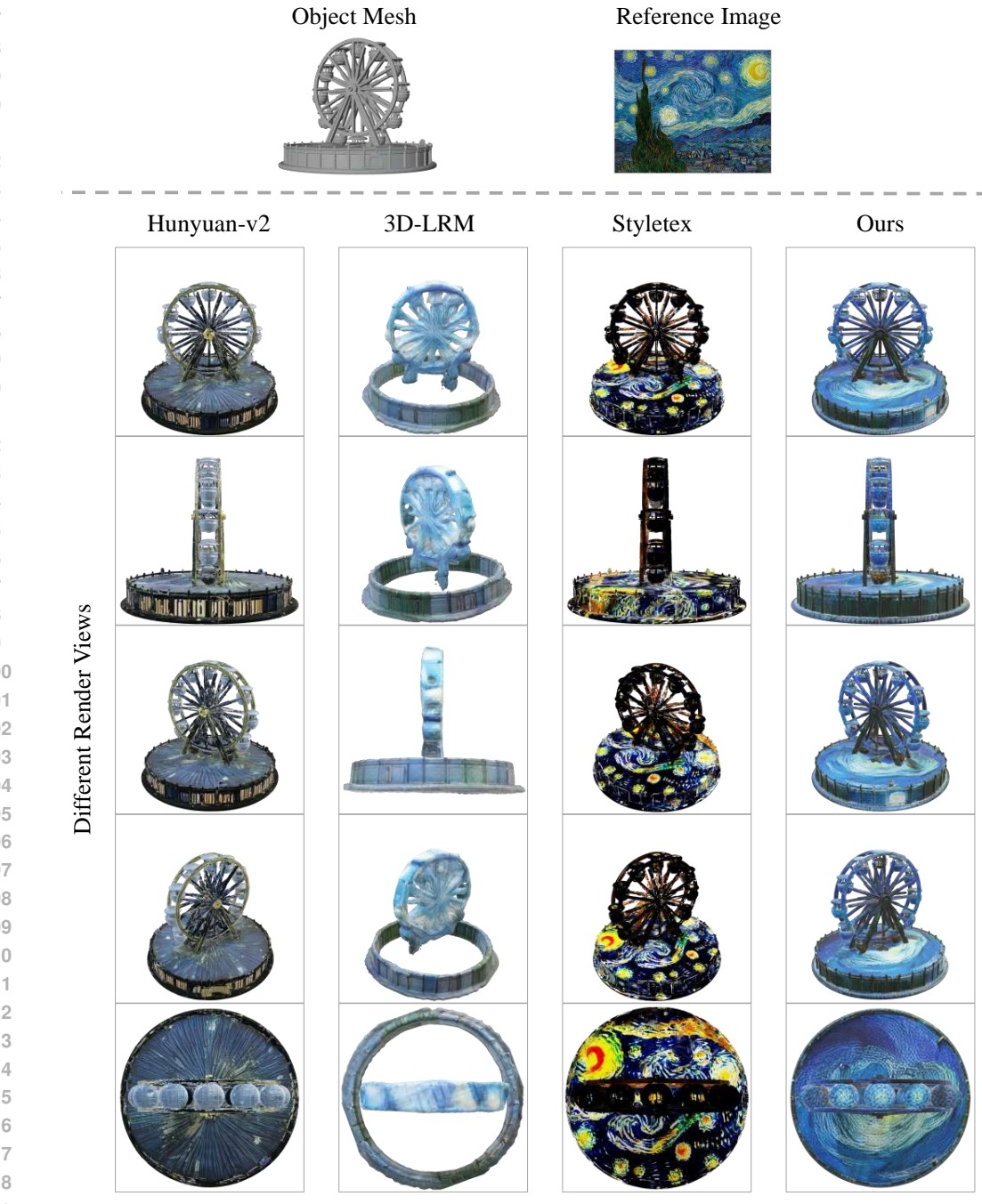

Figure 11: **Qualitative comparison of multi-view stylization results.** Results below the dashed line show outputs from different methods across multiple viewpoints.

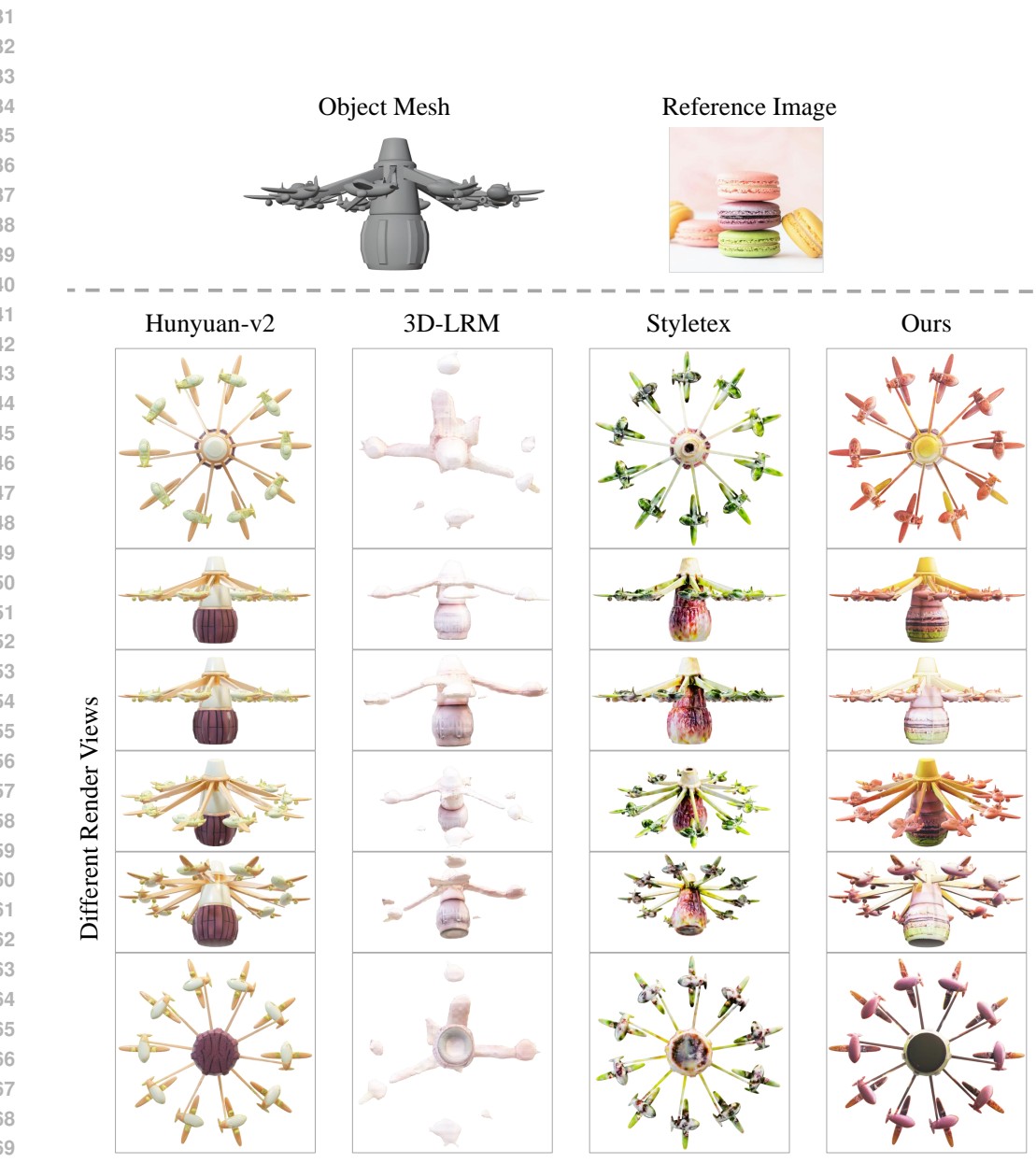

Figure 12: **Qualitative comparison of multi-view stylization results.** Results below the dashed line show outputs from different methods across multiple viewpoints.

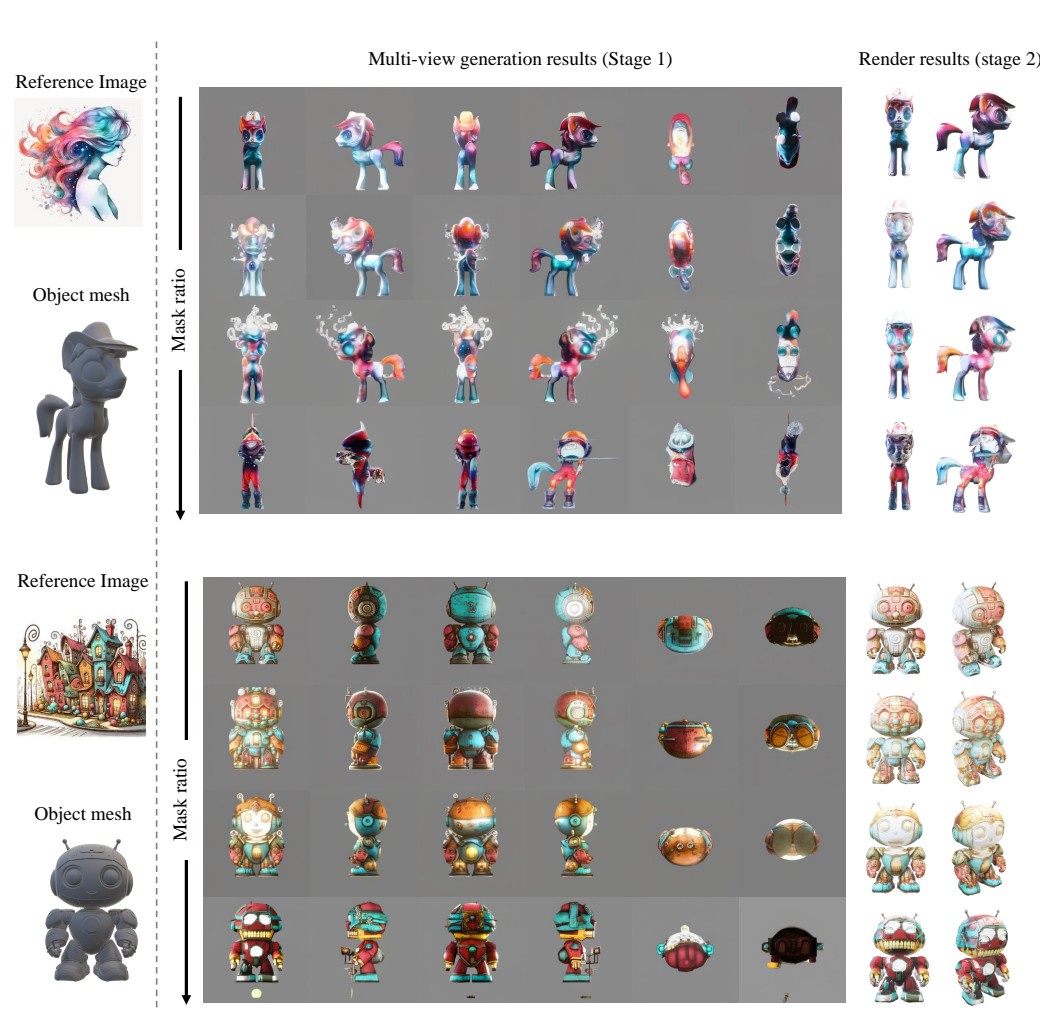

Figure 13: **Ablation study on the mask ratio** $p$**.** This figure illustrates the impact of different mask ratio settings on our two-stage process. The panels on the right side of the dashed line show the multi-view generation results and the final rendered results. Each row corresponds $p$ set to a specific ratio: 0.0, 0.25, 0.5, and 0.75, from top to bottom.

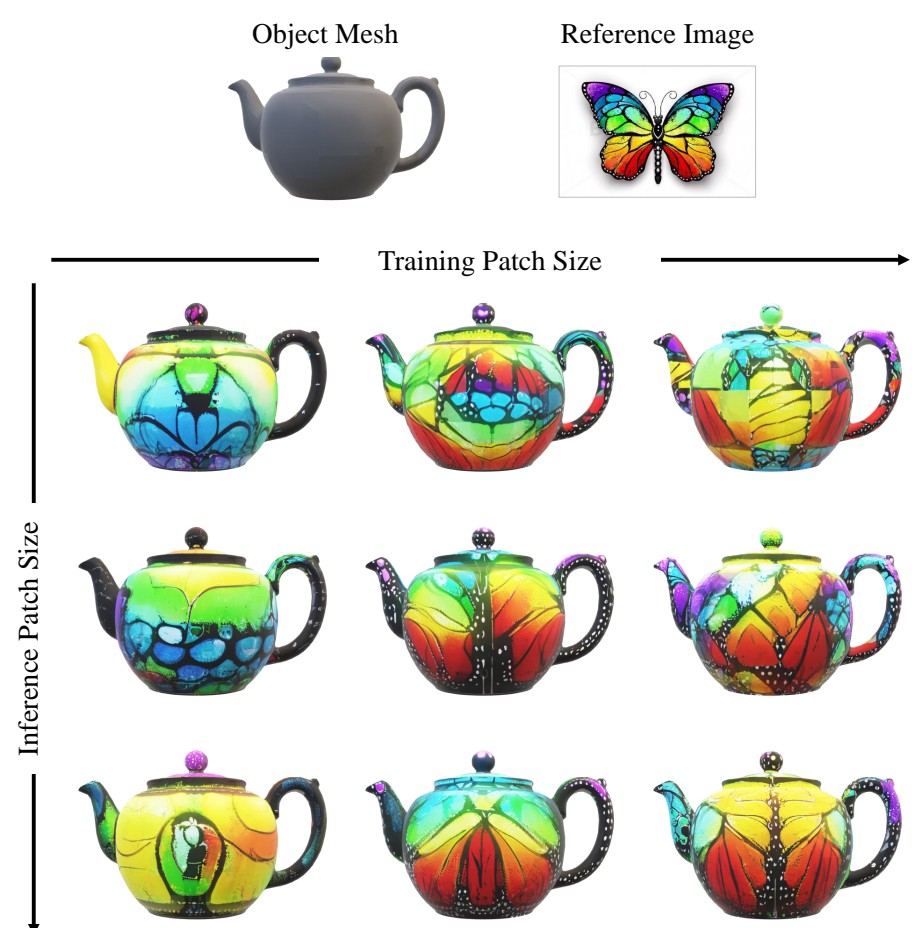

Figure 14: **Ablation study on patch size** $S$**.** We analyze the impact of varying patch sizes during training and inference. The entire image size is fixed at $512 \times 512$, and the image patch size is set to $S \times S$. From left to right, the columns show rendering results using training patch sizes of $32 \times 32$, $64 \times 64$, and $128 \times 128$, respectively. From top to bottom, the rows correspond to inference patch sizes of $32 \times 32$, $64 \times 64$, $128 \times 128$.

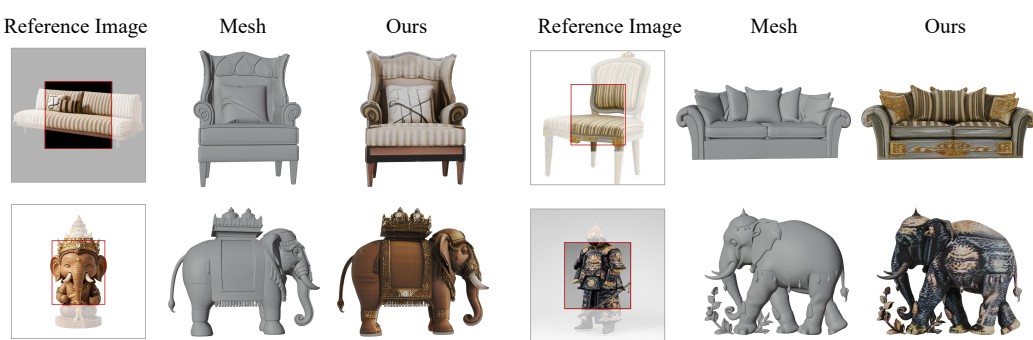

Figure 15: **Partial stylization results.** For reference image, we preserve the partial region (as presented as red box) and mask other regions for partial inference.

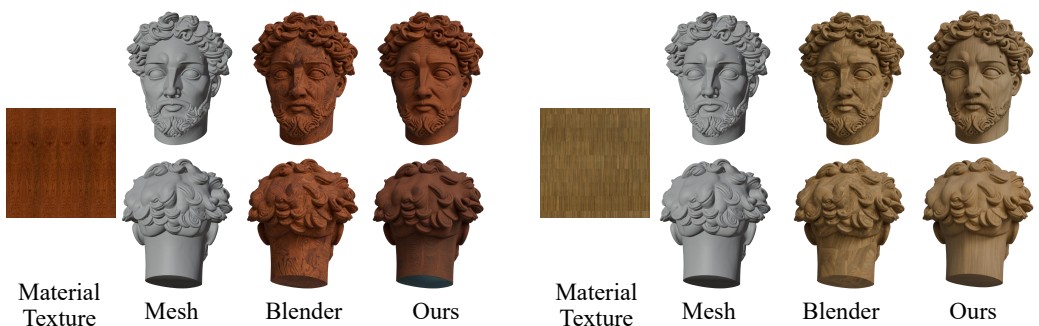

| Material Texture | Mesh | Blender | Ours | Material Texture | Mesh | Blender | Ours |

Figure 16: **Comparison with conventional renderers on tileable texture generation.** Our method effectively eliminates seam artifacts compared to Blender, which often produces inconsistent texture directionality across UV boundaries.

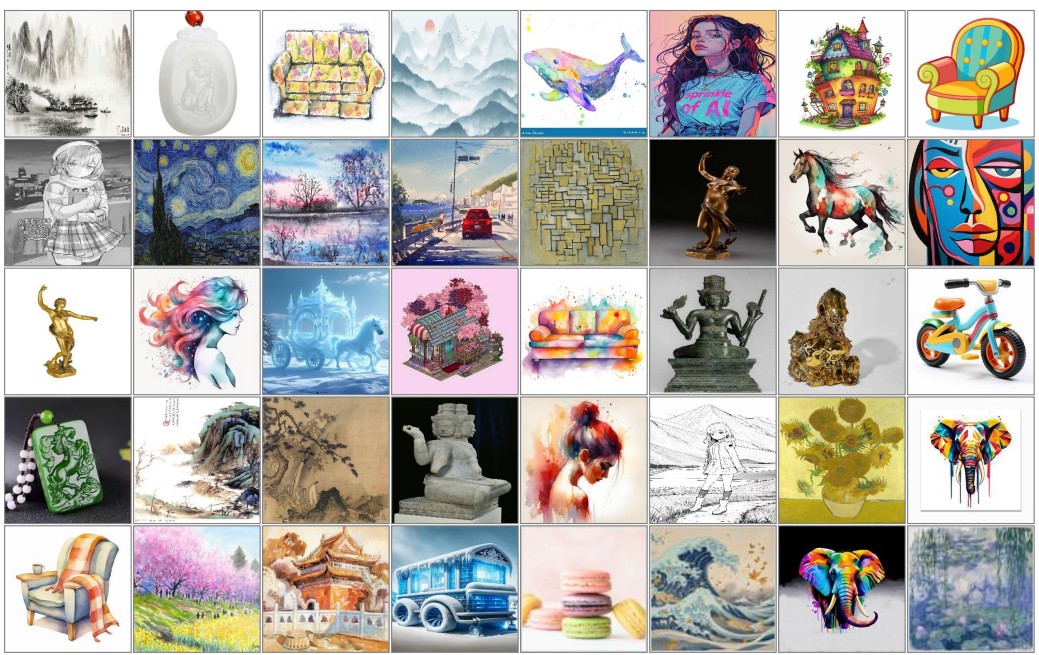

Figure 17: **Visualization on our self-collected reference images.**

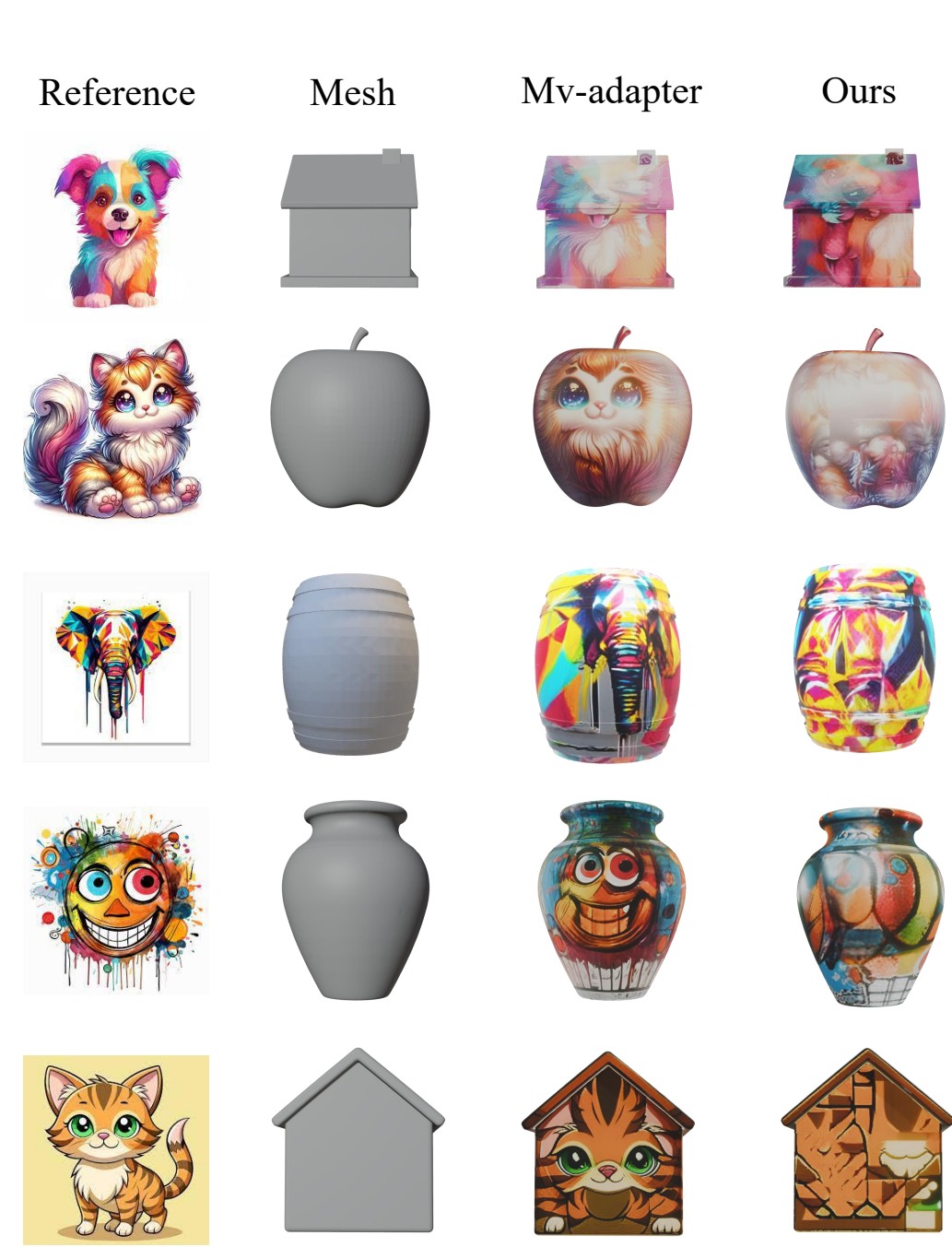

| Reference | Mesh | Mv-adapter | Ours |
|-----------|------|------------|------|

Figure 18: **Disentanglement results between Jigsaw3D and MV-Adapter.** MV-Adapter tends to transfer the entire reference appearance directly to the target mesh, whereas our method successfully isolates and extracts only the style from the reference image.

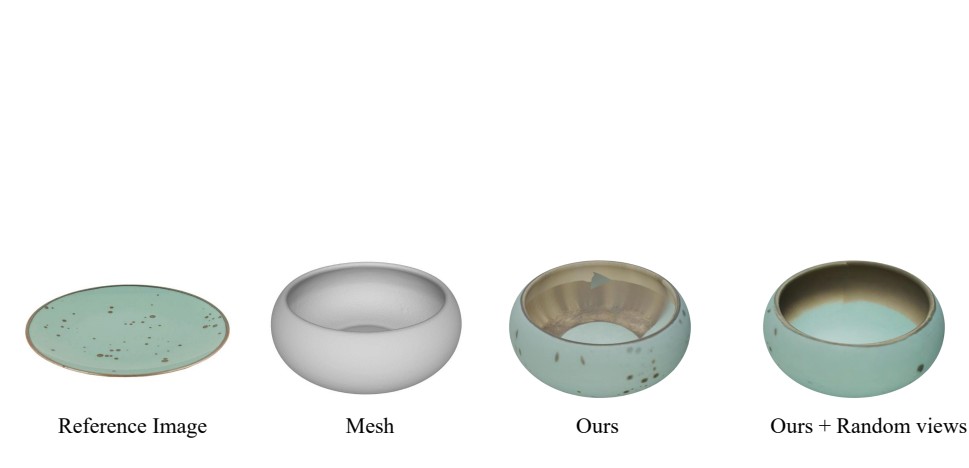

| Reference Image | Mesh | Ours | Ours + Random views |

Figure 19: **Failure Cases in UV Baking and Improvement Strategy.** By generating complementary random views, our method provides more complete texture coverage and effectively addresses missing or obscured areas via robust texture completion.

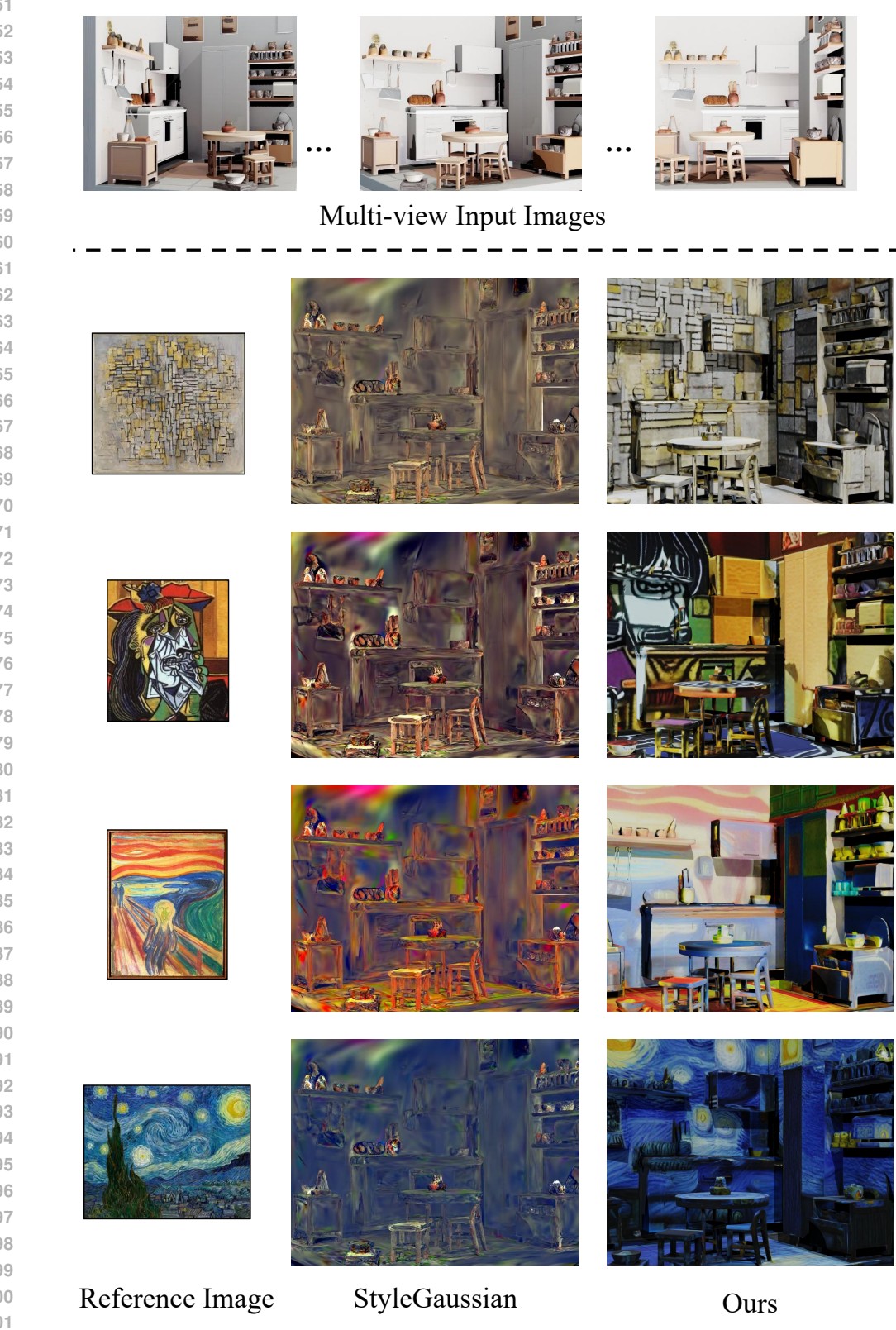

Multi-view Input Images

Reference Image          StyleGaussian          Ours

Figure 20: **Comparative results between our method and 3DGS-based stylization approaches.**

