# OpenReview forum: "Jigsaw3D: Disentangled 3D Style Transfer via Patch Shuffling and Masking"
_ICLR.cc/2026/Conference — Submitted to ICLR 2026_

### Official Review · Reviewer_6UKt · 2025-10-26

**Soundness:** 3
**Presentation:** 3
**Contribution:** 3
**Rating:** 8
**Confidence:** 3

**Summary:**

This paper presents JIGSAW3D, a novel 3D style transfer method to restyle the texture of a 3D assets with a provided reference image.
Specifically, JIGSAW3D operation contains spatial shuffling and random masking of the reference patches. It can suppress object semantics and isolate stylistic statistics.

The main contributions are two folds: (1) Jigsaw-based reference to construct a pseudo-paris dataset for supervised training; similar operation can be applied to the real reference image at inference stage. (2) Training reference-attention module incorporating self, multi-view, and reference attentions to inject disentangled styles to the diffusion model.

Experiments are conducted on Objaverse. Reference images are from both WikiArt dataset and collected 40 extra images. Baselines include three 2025 method for stylisation. Both quantitative and qualitative results are shown. Ablation studies w.r.t. Jigsaw are provided.

**Strengths:**

## Strengths

- 1. The idea of Jigsaw has been explored in feature learning, e.g., Unsupervised Visual Representation Learning [R1], which shows effectiveness in context prediction. This paper leverages this simple but powerful operation, and combines it with masking operation, which is also a helpful operation in self-supervised learning such as MAE [R2]. The Jigsaw3D serves as a simple but effective operation to destroy semantics while preserving style information contained in image patches. Overall, it is a smart and novel approach to disentangle semantic from style.
- 2. A pseudo pair dataset is constructed from Objaverse by generating 6 orthogonal views as ground-truth and 4 random views as reference images. This method enables supervised training but is free from heavy pair annotation.
- 3. Reference attention consists of three attention modules: self-attention, multi-view attention, and reference attention. It leverages the cross-view interaction and inserts reference image styles into the 3D views.
- 4. Experimental results are diverse and evident.
> Quantitative comparisons are conducted with three recent (2025) methods, incicating the better performance and lower cost time.
> Qualitative results are shown on both WikiArt and self-collected reference images.
> Diverse results w.r.t. multi-object, partial, tieable scenes are shown.


[R1] Doersch, C., Gupta, A., & Efros, A. A. (2015). Unsupervised visual representation learning by context prediction. In Proceedings of the IEEE international conference on computer vision (pp. 1422-1430).
[R2] He, K., Chen, X., Xie, S., Li, Y., Dollár, P., & Girshick, R. (2022). Masked autoencoders are scalable vision learners. In Proceedings of the IEEE/CVF conference on computer vision and pattern recognition (pp. 16000-16009).

**Weaknesses:**

## Weaknesses
- 1. Paper oranisation and description of the pseudo pair dataset construction part can be improved. In current version of writing, the training/inference stages are kind of mixed together.
> E.g., in my understanding, during training in Figure 2, I should stand for a random view refence image from the Objaverse. During inference, the reference image is user provided.
> In Sec 3.1 and Sec 3.2, it is also confusing to distinguish the constructed dataset/pseudo reference image and the real user-provided reference image at inference time.
> Moreover, from Fig 2, it seems like during inference of each reference image stylisation, the model is fine-tuned. Is this the case? Or only during training on the pseudo paired dataset, the model is finetuned and keep frozen at inference time?
- 2. In Fig. 3, as Divisions increases, Gram Matrix increases significantly. While in Tab. 1, Gram is a metric expected to be the smaller the better. How to understand this inconsistency?
- 3. Why the jigsaw patch sizes are different during training and inference?
- 4. Will the self-collected 40 reference images be released? This will facilitate future comparison.
- 5. The proof in A.2 is trivial.

**Questions:**

The questions are provided in the above Weaknesses points 1. ~ 4.

---

> ### Author Response · Authors · 2025-11-20
> **Official Comment by Authors**
>
> We are grateful to the reviewers for your insightful feedback and thorough evaluation. The suggestions have been invaluable in strengthening our work. We try to address all your concerns below.
>
> **Weakness 1: Unclear description of pseudo pair dataset construction**
>
> Thank you for these valuable suggestions. During training, the reference image is indeed sampled from Objaverse, while during inference, it is provided by users. Our model is trained only on the pseudo-paired dataset and remains frozen during inference. We will improve these descriptions in Sec 3.1 of the revised version.
>
> **Weakness 2: On the inconsistency of Gram evaluation between Fig. 3 and Table 1**
>
> Thank you for this insightful observation. The two results are not contradictory, as both use Gram values to represent style fidelity. Note that the shuffling operation still affects the style because if we shuffle every pixel, then the style is also lost.  Figure 3 aims to illustrate the trade-off: as the number of divisions increases, the Gram value experiences a moderate and expected rise due to shuffled style encoding, while the semantic content from the reference decreases significantly. This observation guides us to select an optimal division number that achieves a balance between style fidelity and semantic preservation. These results do not contradict Table 1, where lower Gram values indicate better style matching under consistent division settings across methods.
>
> **Weakness 3: On different patch size settings in training and inference**
>
> Thank you for this question. The difference in jigsaw patch sizes between training and inference is motivated by the inherent bias in our dataset. Specifically, objects in our training data are relatively small, which encourages a finer-grained patch division to better learn localized representations. During inference, however, the input style images generally occupy the full image and are larger in scale, and thus a correspondingly coarser patch division is applied to avoid excessive fragmentation. The ablation study on patch size settings (provided in **Appendix A.5 and Fig. 14**) demonstrates that this specific configuration produces the most stable and robust results.
>
> **Weakness 4: On the release of self-collected reference images**
>
> Thank you for the suggestion. These 40 self-collected reference images are now provided in **Appendix Fig. 17**.

---

> > ### Comment · Reviewer_6UKt · 2025-11-26
> >
> > Thank the authors for the reply!
> > The weaknesses points 1-4 have been addressed in the rebuttal.
> > Point 5 is a comment not a criticism because it is not stated as a separate contribution in this paper.

---

### Official Review · Reviewer_5AzY · 2025-10-30

**Soundness:** 2
**Presentation:** 3
**Contribution:** 2
**Rating:** 2
**Confidence:** 4

**Summary:**

This paper introduces JIGSAW3D, a novel framework for disentangled 3D style transfer that transfers artistic styles from 2D reference images to 3D meshes while preserving object identity and multi-view consistency. The core innovation is a Jigsaw transform that disentangles style from content by randomly shuffling and masking non-overlapping image patches, destroying global semantics while preserving local style statistics (e.g., mean, variance). This enables scalable synthesis of pseudo-paired style-texture training data from existing textured 3D assets without curated style datasets. The authors propose a multi-view diffusion model conditioned on disentangled style features and geometric cues (normals/positions). A key component is the Reference Attention mechanism, which dynamically injects style features into the denoising U-Net alongside self-attention and multi-view attention blocks. Stylized multi-view outputs are baked into UV textures via visibility-aware reprojection and seam-aware blending.

**Strengths:**

1. Originality. This paper presents a genuinely creative formulation of the 3D style disentanglement problem. The proposed jigsaw transform adapts the idea of patch shuffling—originally from 2D stylization—to the 3D domain in a surprisingly effective way. By disrupting spatial coherence while preserving local statistics, the authors successfully synthesize pseudo-paired style–texture data from unpaired assets (e.g., Objaverse), providing an elegant workaround for the lack of curated 3D style datasets. The reference attention mechanism—which combines self-, multi-view, and style-guided cross-attention—is both principled and practical. While its components are not entirely new, their integration for disentangled 3D style transfer feels original and well-motivated.

2. Quality. The method is technically solid and well-executed. The jigsaw operation (Eq. 1–2) convincingly preserves style statistics while removing semantics, and the inclusion of geometric cues through normal and position maps strengthens structural fidelity. The multi-view diffusion pipeline is coherent and thoughtfully designed, and the final texture baking—featuring visibility-aware reprojection and UV inpainting—demonstrates strong attention to practical details. The experiments are thorough, both visually compelling and quantitatively convincing. Overall, the technical quality is high and the implementation appears carefully considered.

3. Clarity. The paper is very clearly written. The motivation is intuitive and well-argued, and the methodology section flows logically from data preparation to generation and baking. Figures and equations are well integrated into the narrative, enabling readers to grasp the key ideas efficiently. The ablation and limitation sections are appropriately placed and easy to follow. It is one of the clearest papers in this area.

4. Significance. The potential impact of this work is notable. The ability to perform scalable, multi-object style transfer without per-asset optimization directly addresses a major bottleneck in 3D content creation, with immediate applications in gaming, VR, and digital art pipelines. The jigsaw-based disentanglement strategy may also inspire future research beyond stylization, such as in texture synthesis or domain adaptation. The authors’ plan to release code further enhances the potential influence of this work.

**Weaknesses:**

1. Novelty Overstated; 2D-to-3D Adaptation Not Fully Justified. The proposed jigsaw transform—combining patch shuffling and masking—closely resembles ideas from the 2D style transfer literature, where positional embeddings are shuffled for disentanglement. While the adaptation to 3D is interesting, the paper does not clearly explain why spatial patch shuffling provides a unique advantage for preserving geometry compared to feature-space methods. Suggested Improvement: Include experiments that compare the jigsaw-based disentanglement with feature-space alternatives such as Gram or AdaIN-based approaches in a 3D setting. It would also help to quantify geometric distortion (for example, using Chamfer distance) to justify the necessity of the jigsaw operation.

2. Inadequate Evaluation of Geometric Fidelity. The paper claims to maintain high geometric fidelity, but no quantitative evaluation is provided. The visual examples are persuasive, yet they are insufficient for complex geometries or subtle distortions. Suggested Improvement: Add geometric metrics—such as normal consistency or Chamfer distance—to objectively assess the preservation of shape. Testing on geometrically challenging assets (e.g., fractal or thin structures) would also strengthen the claims.

3. Insufficient Ablation of Core Components. Several critical design choices are not thoroughly validated. In particular, the necessity of the jigsaw transform is not tested against an unshuffled baseline, and the contribution of the three attention branches (self, multi-view, and reference) is not individually analyzed. Suggested Improvement: Provide ablations that remove key components—such as using the raw reference image without shuffling, or disabling the reference attention—to show their quantitative effect on style disentanglement and consistency.

4. Limited Validation of Multi-Object Stylization. The multi-object stylization results look promising, but the evaluation remains qualitative. It is unclear how consistent the style remains across objects within the same scene, or how geometry coherence is maintained when multiple shapes are involved. Suggested Improvement: Introduce scene-level measurements such as inter-object style similarity or surface normal alignment. Testing in more complex, cluttered environments would better demonstrate scalability and robustness.

5. Baking Process Robustness Not Fully Explored. The texture baking process, including seam-aware blending and UV inpainting, is only briefly discussed. Potential failure cases such as heavy occlusion or non-manifold geometry are not examined, and no quantitative UV-space evaluation is given. Suggested Improvement: Show examples of baking failures and describe how the system handles them. Reporting UV-space metrics like seam error or texture continuity would provide stronger support for the robustness of the method.

6. Efficiency Claims Unsubstantiated. The paper emphasizes scalability without per-asset optimization but does not include timing benchmarks. Without concrete measurements, it is difficult to assess whether the method is truly more efficient than prior optimization-based approaches. Suggested Improvement: Report training and inference time per object, and include comparisons with standard diffusion or optimization-based baselines to demonstrate actual runtime improvements.

**Questions:**

1. Necessity of Spatial Patch Shuffling. Could the authors clarify why spatial patch shuffling is essential for 3D style disentanglement? Similar effects have been achieved in 2D through feature-space shuffling. A comparison with a latent-space shuffling baseline would help justify this design choice.

2. Quantitative Evaluation of Geometric Fidelity. The paper emphasizes preserving geometry but relies mainly on visual comparisons. Could the author provide quantitative metrics such as Chamfer distance or normal-based errors between the original and stylized meshes?

3. Ablation on Jigsaw and Reference Attention. Can the authors include ablations to show (a) the effect of removing the jigsaw operation and (b) the contribution of each attention branch (self, multi-view, reference)? Quantitative results would make these components’ roles clearer.

4. Generalization to Complex Geometries. How does the method perform on objects with fine details, thin structures, or non-manifold geometry? Some examples or metrics on such challenging cases would strengthen the generalization claim.

5. Efficiency and Scalability. Since the paper highlights efficiency compared to per-asset optimization, could you report inference time, training cost, and memory usage?

6. Interpretation of Masking in the Jigsaw Operation. What role does stochastic masking play in style transfer? Does it mainly promote robustness or feature diversity? Results with different mask ratios could clarify its effect.

7. Failure Cases in UV Baking. Could the paper discuss typical failure cases in the baking process and how they are mitigated? A few visual examples would improve transparency.

---

> ### Author Response · Authors · 2025-11-20
> **Official Comment by Authors**
>
> We thank the reviewer for your feedback. We carefully consider all the comments and address the Question point-by-point below.
>
> **Question 1: On the Necessity of Spatial Patch Shuffling**
>
> We appreciate the reviewer's input. Patch shuffling is essential for 3D adaptation, as it fundamentally differs from 2D generation. Specifically, we employ jigsaw-based patch shuffling and masking operations to construct a pseudo style-texture paired dataset. This approach effectively guides multi-view style-consistent generation and facilitates UV projection, thereby enabling robust 3D adaptation.
>
> **Question 2: On the Quantitative Evaluation of Geometric Fidelity**
>
> We respectfully clarify that our task focuses on stylization, and our experiments therefore primarily evaluate reference style fidelity and semantic preservation. Since all baseline methods utilize the same input mesh, a quantitative comparison of geometric fidelity is less critical in this context. We will report this in the revision.
>
> **Question 3:  On Insufficient Ablation of Core Components**
>
> We have provided an ablation study on the jigsaw operation in Fig. 5\. However, removing either the multi-view or reference attention module is infeasible for conducting a meaningful ablation study, as both modules must operate jointly to enable the core processes of multi-view generation and 3D stylization.
>
> **Question 4:  Generalization to Complex Geometries**
>
> Thanks for your suggestion. We have considered generalization to complex geometries, which is provided in fig. 8, fig. 9, and fig. 10\. The results show successful stylization on structurally complex shapes.
>
> **Question 5: On Efficiency and Scalability**
> As indicated in Table 1, we have reported the inference time for our method. Regarding training cost, the model was trained for 5 days on 8 GPUs (each delivering up to 312 TFLOPS in FP16 precision) using the 8w Objaverse data, with a memory consumption of 40GB.
>
> **Question 6: Interpretation of Masking in the Jigsaw Operation**
>
> Thank you for your feedback. We provide this in fig. 13, which shows appropriate masking in the jigsaw operation helps enhance the diversity of learned feature representations.
>
> **Question 7: Failure Cases in UV Baking**
>
> Thanks for your input. We acknowledge that severe occlusion may occasionally lead to incomplete texture baking in certain regions. To address this, we employ a multi-view completion strategy by randomly selecting additional camera viewpoints to fill in the missing areas. **Fig. 19** demonstrates the effectiveness of this approach.

---

### Official Review · Reviewer_iYfU · 2025-10-31

**Soundness:** 3
**Presentation:** 3
**Contribution:** 2
**Rating:** 4
**Confidence:** 4

**Summary:**

This work proposes Jigsaw3D, a method for generating style-image conditioned textures for meshes. A key observation of this work is that existing datasets conflate semantics and style within the textures of their meshes. To address this, Jigsaw3D proposes a texture processing step that spatially shuffles and randomly masks patches of the reference textured renders. This allows the resulting patches to be independent of the object semantics and isolates the style component of the texture. Using these "jigsawed" patches, this method is able to train a multi-view diffusion model that is conditioned on cross-attention with these patches, enabling style-only conditioned multi-view diffusion. The method is evaluated both qualitatively and quantitatively, reporting improved results over baseline methods.

**Strengths:**

- This work identifies an issue with current image guided texturing approaches (style and semantics are very interconnected) and proposes a method that solves this problem, providing an original contribution to the community.
- The proposed “jigsaw” pre-processing step enables the creation of a dataset of style jigsaw  images and textured mesh pairs from a dataset of just textured meshes. This is a creative solution to the problem of disentangling style from semantics and can be used by followup works aiming to separate style and semantics.
- Jigsaw3D’s training-based approach leads to faster run times than per-shape optimizations using score distillation and is thus more practical than distillation methods.

**Weaknesses:**

- The desired degree to which semantics of the input mesh should be preserved seems very subjective. In image guided texturing there is a tradeoff between preserving the semantics on the input mesh and adhering to the style of the reference image. Jigsaw3D seems to preserve more semantics in some examples and less in others. For example, in Fig. 4, the lamp in StyleTex colors the shade with a more traditional paper texture while the base is colored bronze and the Jigsaw3D result colors the entire thing bronze. Some might argue that the full bronze texture has more fidelity to the reference image while the bronze base and paper shade has better semantic preservation of the input shape. However in the same figure, for the example with the pink floral dress on the car, the Jigsaw3D has more semantic preservation and less fidelity to the reference image. From these examples it appears that StyleTex and Jigsaw3D are comparable qualitatively, and it’s not clear what the desired behavior in these cases should be.
- In some cases (bronze lamp, cartoonish houses on bus), the results from MVAdapter appear comparable to Jigsaw3D and MVAdapter also has a similar fast runtime. Thus from the qualitative results, it is not clear that Jigsaw3D performs better than baseline methods.
- Given that a main claimed contribution of this work is the disentanglement of style and semantics I would expect more experiments explicitly showing this. Tab. 1 reports Gram and AdaIN metrics, but why these capture style / semantics disentanglement is not explained.

**Questions:**

See weaknesses.

---

> ### Author Response · Authors · 2025-11-20
> **Official Comment by Authors**
>
> We are grateful to the reviewers for your thorough evaluation and valuable suggestions. We hope our reply addresses all your concerns.
>
> **Weakness 1: On evaluation trade-off and comparison with StyleTex**
>
> Thank you for your feedback. We agree that a successful stylization should preserve the semantic structure of the input mesh while faithfully matching the style of the reference image. For the lamp case in Fig. 4, our method generates a whole golden texture because the given style image is golden. The result produced by StyleTex for the lamp case exhibits noticeable color deviation and unrealistic hallucination of stripes, which are not aligned with the underlying geometry. In contrast, our method accurately captures the material of the reference and maintains the structural semantics of both the lamp base and the frame. Similarly, in row 3 (the car example), our approach preserves the semantic structure of the car windows while effectively reflecting the colors and aesthetics of the reference style. Other examples further demonstrate the balanced performance of our method in maintaining structure and style, such as the sunflowers on the roof in row 2 and the roof tiles in row 6\.
>
> **Weakness 2: Clarifying the advantage of Jigsaw3D over MVAdapter**
>
> Thank you for raising this point. Our observations indicate that MVAdapter tends to directly project the entire reference image onto the mesh surface. For instance, in Fig. 4 row 1, it incorrectly applies the white background onto the lamp, and in row 5, it misplaces window-like patterns onto the bus body. We also provide further comparisons **(Appendix Fig. 18\)** to show that our method effectively disentangles style from semantic structure.
>
> **Weakness 3: The need for more experiments and metrics explanation on disentanglement**
>
> We agree that further experiments of disentanglement would strengthen the paper.  We add **Appendix Fig. 18** to demonstrate our superiority in disentangling irrelevant semantic content. Regarding quantitative metrics, we employ Gram and AdaIN metrics (detailed in **Appendix A.3**) to capture style similarity, while using CLIP metrics to evaluate content preservation. Our strong performance across both style and content metrics collectively demonstrates effective style-content disentanglement.

---

### Official Review · Reviewer_pTPC · 2025-10-31

**Soundness:** 3
**Presentation:** 3
**Contribution:** 2
**Rating:** 4
**Confidence:** 4

**Summary:**

This paper introduces a new 3D texture generation framework that isolates the color palettes, strokes, and textures from the style image and maps them to the surface of a given 3D mesh/asset. This framework consists of a trained diffusion model for multi-view stylized image generation and a 3D texture baking process to project the stylized views into UV space.

The major contribution of this paper is the introduction of the "jigsaw" operation, which involves random shuffling and masking of image patches to destroy the semantic structure while preserving the style statistics, allowing supervised training. Experiments show better style fidelity and cross-view consistency compared with StyleTex, MV-Adapter, and 3D-Style-LRM, with competitive runtime.

**Strengths:**

1. The design of patch-shuffle + mask achieves style-content disentanglement theoretically and empirically. Quantitative analysis and mathematical proof are provided to demonstrate the effectiveness and functionality of this design in terms of style-semantic decoupling and mean/variance preservation. Additionally, this method enables supervised learning in a clean and scalable manner.

2. Even though the multi-branch attention and reference-attention formulation are not very novel, they have been demonstrated to be simple and effective for this task, which is valuable for future study.

3. The writing of this paper is clear and logically structured, with professional schematic diagrams and comparison figures.

**Weaknesses:**

1. The scope of this topic is relatively weak, focusing on the texture stylization of 3D assets. Recent 3D style transfer works can be applied at both the object and scene levels. Can this method be transferred to general 3D style transfer? Can the authors compare this method with recent 3DGS-based style transfer models (by giving six or more views to these models)?
2. Patch shuffling has already been studied in **StyleAdapter** to reduce the semantic information. What is the difference between your observation and theirs.
2. No perceptual or human evaluation of realism and consistency is provided.
3. Only 20 test meshes are used for evaluation, which may be too small to demonstrate the generalization ability of the proposed methods.

**Questions:**

1. Based on the content of this paper, I suggest that the authors change the title to "3D Texture Generation" instead of "3D Style Transfer," as the latter is too general and not suitable for this paper.
2. If more comparisons with recent 3DGS-based methods can be provided, I would like to increase the score accordingly.

---

> ### Author Response · Authors · 2025-11-20
> **Official Comment by Authors (1/2)**
>
> Thank you for the detailed and constructive feedback. We treasure the opportunity to address your concerns and improve our work.
>
> **Response to Weaknesses:**
>
> **Weakness 1: Provide scene-level style transfer application and comparison with 3DGS-based models**
>
> We agree that general 3D style transfer represents a broad and valuable research direction. Our method mainly focues on the object-level stylized texture generation and has the potential to extend to scene-level stylization. We have demonstrated the ability on the object-level and include additional experiments for the scene-level. For scene-level stylization, we have conducted a comparisons with StyleGaussian \[1\], with newly added results provided in **Appendix Sec. A.10 and Fig. 20**. In terms of experimental setup, for our method, we first send 3 different input views to generate a scene-level mesh using Hunyuan3D. The mesh and reference image are then processed by our Jigsaw3D framework and finally render the stylized scene images. For 3DGS stylization, we provide 120 multi-view images to StyleGaussian for point cloud reconstruction and Gaussian splatting stylization. Note that Fig.20 shows our method is only trained on object level meshes but already shows reasonable performances for a 3D mesh scene.  Due to the time limits, we only conduct a direct experiment but will include more scene stylization results in the revision.
>
> **Weakness 2: Clarifying differences from prior patch shuffling methods**
>
> Thank you for raising this important point. We acknowledge that patch shuffling has been widely explored as a common operation in many tasks and StyleAdapter uses it for stylized image generation. However, our approach introduces three key differences: First, we are the first to apply this technique in the more challenging domain of 3D stylized mesh generation while StyleAdapter only uses it for 2D image stylization. Second, while prior methods like StyleAdapter primarily shuffle position embeddings with a fixed patch size, our core contribution lies in using a patch-level shuffling with varying patch sizes and we additionally add a masking strategy to construct style-texture pairs for training our multi-view diffusion model. Finally, we provide a thorough empirical analysis (Fig. 3\) that explores the relationship between shuffle intensity and style-content disentanglement—an important quantitative observation not established in previous methods. We have demonstrated the effectiveness of our method, and we believe that our work could inspire future 3D stylization methods.
>
> **Weakness 3:  Lack of Perceptual or Human Evaluation**
>
> Thanks for your suggestions. We conducted a user study with 20 participants to evaluate style transfer quality. Participants rated the results based on view consistency of the 3D assets, overall success of reference style transfer, and preservation of original structural semantics, using a 0–100 scale for each criterion. The final score was averaged across all evaluators and aspects. As shown below, our method achieves a clear superiority in user preference:
>
> | Method | User Score (0-100) |
> |--------|-------------------------------|
> | StyleTex | 72.3 |
> | MV-Adapter| 76.0 |
> | 3D-style-LRM | 61.8 |
> | **ours** | **82.4** |
>
> In addition, regarding our Gram matrix and AdaIN evaluation metrics, we clarify that these are designed to serve as perceptual evaluation metrics (we include more details in **Appendix A.3**).
>
> **Weakness 4: Limited number of test meshes (20) used for evaluation.**
>
> We consider 20 test meshes with 40 reference styles, which results in 800 stylized generation for evaluation. Due to the high computational cost of SDS-based baselines, inference over 20 meshes and 40 reference styles already requires several days.
>
> Following your suggestions, we conducted additional experiments, with 10 representative meshes from objaverse and our self-collect reference image, and report the full results below:
>
> | Method | Gram ↓ | AdaIN ↓ | CLIP ↓ |
> |--------|--------|---------|--------|
> | StyleTex | 5.42 | 126.38 | **0.206** |
> | MV-Adapter| 4.83 | 115.27 | 0.213 |
> | 3D-style-LRM | 5.65 | 137.45 | 0.211 |
> | **ours** | **4.78** | **112.91** | 0.210 |
>
> We will include these additional comparisons in the revised version to further strengthen the evaluation.
>
> \[1\] StyleGaussian: Instant 3D Style Transfer with Gaussian Splatting, SIGGRAPH Asia, 2024

---

> > ### Author Response · Authors · 2025-11-20
> > **Official Comment by Authors (2/2)**
> >
> > **Response to Questions:**
> >
> > **Question 1: Changing the title to "3D Texture Generation"**
> >
> > Thank you for this constructive suggestion. We will revise the paper's title to "3D Stylized Texture Generation" to better reflect our work's scope.
> >
> > **Question 2: Requiring comparisons with recent 3DGS-based methods**
> >
> > Thank you for the suggestion. Please refer to our response to **Weakness 1,** we provide new comparisons in **Appendix Sec. A.10 and Fig. 20**.

---

### Author Response · Authors · 2025-11-30
**Global Reply to Reviewers' Comments**

We sincerely appreciate all reviewers for their valuable feedback. Here we present our responses and the corresponding modifications designed to fully address the reviewers' concerns:

1.  **For Reviewer pTPC:** We have added Human Evaluation results, expanded test meshes evaluation, and included comparisons with 3DGS-based methods (see Appendix Sec. A.10 and Fig. 20). We believe the comparisons with 3DGS-based methods demonstrate our method's capability to extend to scene-level stylization, thereby addressing the reviewer's concern. **The reviewer's comment suggests a score increase upon inclusion of these results.**

2.  **For Reviewer iYfU:** We have provided additional disentanglement experiments (see Appendix Fig. 18) that further demonstrate our advantages over existing approaches. **We believe these results resolve the reviewer's concern about the experimental comparisons.**

3.  **For Reviewer 5AzY:** We have provided more detailed explanations and supplementary descriptions throughout the paper, addressing the reviewer's concerns.

4.  **For Reviewer 6UKt:** We have released our self-constructed dataset (see Appendix Fig. 17). **We thank the reviewer for confirming that our responses satisfactorily address their concerns.**

---

### Meta-Review · Area_Chair_5nZt · 2026-01-07

**Summary:**

Reviewers generally found the idea of using a jigsawed reference to suppress semantics interesting, but the paper did not convincingly clear the “ICLR bar” on empirical strength. In particular, one reviewer noted that the qualitative results often look comparable to baselines and that it is “not clear” Jigsaw3D is better from those comparisons. Others questioned whether the evaluation was sufficiently broad (e.g., limited meshes) and whether the paper’s scope/claims (e.g., beyond object-level) were adequately supported. Given that the strongest claims hinge on disentanglement and practical quality improvements, the evidence felt incremental rather than clearly state-of-the-art.

**Reviewer Concerns:**

Addressed in the rebuttal (partially):
(1) Added a small human study (20 participants) and reported user scores.
(2) Clarified what Gram/AdaIN/CLIP are intended to measure and added additional disentanglement comparisons (Appendix Fig. 18 per rebuttal).
(3) Added a 3DGS-based comparison for scene-level stylization, but framed as a limited “direct experiment.”
(4) Expanded evaluation with additional meshes beyond the original 20 (as reported in rebuttal).

Still outstanding:
(1) Even with clarifications, the main concern about result strength remains: qualitative gains over strong baselines are not consistently obvious, and the “desired behavior” trade-off is still subjective.
(2) The scene-level evidence is preliminary (authors explicitly note time-limited experimentation).
(3) Some requests for deeper component-level validation remain unresolved (e.g., reviewer asked for attention-branch ablations; authors state removing modules is infeasible for a “meaningful” ablation).
(4) Reviewer 5AzY’s calls for geometry-related metrics were not substantively met (authors argue it is less critical since geometry is unchanged across methods).

**Reviewer Scores:**

pTPC (4 → maybe 5): Likely a modest increase, since they explicitly said they would raise the score if 3DGS comparisons were added, and rebuttal added such comparisons plus human eval.
iYfU (4 → 4): The rebuttal addresses their request for clearer disentanglement support, but their core skepticism was about qualitative superiority being unclear, so I expect limited movement.
5AzY (2 → 2): Likely unchanged; multiple requested validations (e.g., geometry metrics, attention ablations) remain unconvincing or were declined as infeasible/“less critical.”
6UKt (8 → 8): Likely unchanged; the reviewer explicitly stated weaknesses 1–4 were addressed in rebuttal.

---

### Decision · Program_Chairs · 2026-01-26

Reject